# Learnable edge detectors can make deep convolutional neural networks more robust

**Jin Ding**[1,2]*, **Jie-Chao Zhao**[1], **Yong-Zhi Sun**[1], **Ping Tan**[1], **Jia-Wei Wang**[1], **Ji-En Ma**[3], **You-Tong Fang**[3]

**1** School of Automation and Electrical Engineering & Key Institute of Robotics of Zhejiang Province, Zhejiang University of Science and Technology, Hangzhou, China, **2** State Key Laboratory of Fluid Power and Mechatronic Systems, Zhejiang University, Hangzhou, China, **3** School of Electrical Engineering, Zhejiang University, Hangzhou, China

* jding@zust.edu.cn

## Abstract

Deep convolutional neural networks (DCNNs) are vulnerable to examples with small perturbations. Improving DCNNs' robustness is of great significance to the safety-critical applications, such as autonomous driving and industry automation. Inspired by the principal way that human eyes recognize objects, i.e., largely relying on the shape features, this paper first designs four learnable edge detectors as layer kernels and proposes a binary edge feature branch (BEFB) to learn the binary edge features, which can be easily integrated into any popular backbone. The four edge detectors can learn the horizontal, vertical, positive diagonal, and negative diagonal edge features, respectively, and the branch is stacked by multiple Sobel layers (using learnable edge detectors as kernels) and one threshold layer. The binary edge features learned by the branch, concatenated with the texture features learned by the backbone, are fed into the fully connected layers for classification. We integrate the proposed branch into VGG16 and ResNet34, respectively, and conduct experiments on multiple datasets. Experimental results demonstrate the BEFB is lightweight and has no side effects on training. And the accuracy of the BEFB-integrated models is better than the original ones when facing white-box attacks and black-box attack. Besides, BEFB-integrated models equipped with the robustness enhancement techniques can achieve better classification accuracy compared to the original models. The work in this paper for the first time shows it is feasible to enhance the robustness of DCNNs through combining both shape-like features and texture features.

## Introduction

It is well known that deep convolutional neural networks (DCNNs) can be fooled by examples with small perturbations [1–6], which poses potential hazards for safety-critical applications, e.g., autonomous driving, airport security, and industry automation. Therefore, it is

**Funding:** This work was supported by National Key Research and Development Program of China(2024YFB2409200), Open Foundation of the State Key Laboratory of Fluid Power and Mechatronic Systems(GZKF-202329), and National Natural Science Foundation of China(52293424, 52477065). The funders played no role in the study design, data collection and analysis, decision to publish, or preparation of the manuscript.

**Competing interests:** The authors have declared that no competing interests exist.

of great significance to improve the robustness of DCNNs. Figs 1 and 2 show a clean example and its adversarial counterpart. Adversarial example (AE for short) in Fig 2 is with small noises which can induce DCNNs to make wrong decisions. However, these noises can be filtered by human eyes easily. Researches show that the principal way human beings recognize the objects is largely relying on the shape features [7,8], and that's why human beings are not susceptible to the noise imposed in Fig 2. Inspired by this, it is natural to ask, is it possible to make the DCNNs more robust by learning the shape-like features?

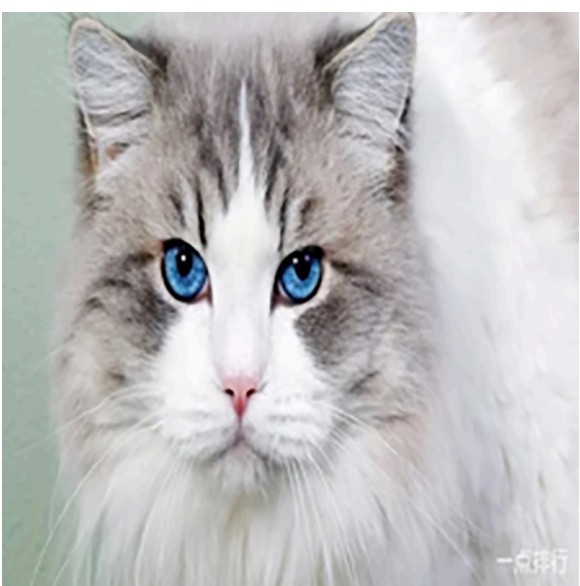

**Fig 1. Clean example.**

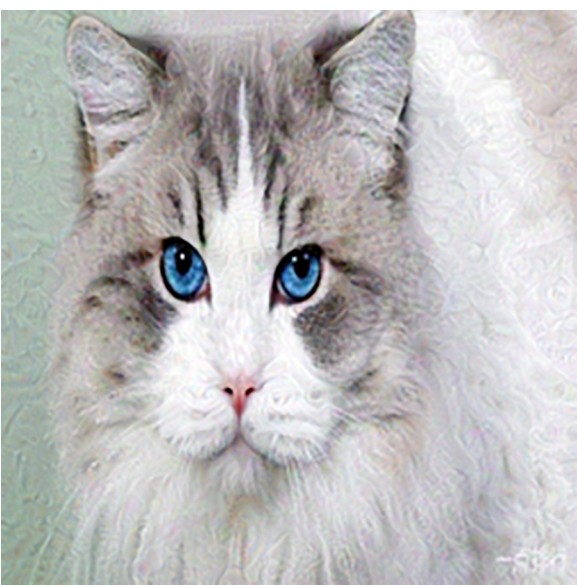

**Fig 2. Adversarial example.**

Fig 3 and Fig 4 shows the thresholded edge images of Fig 1 and Fig 2, respectively. From the figures, it can be observed that the thresholded edge images for Figs 1 and 2 are almost the same, which indicates the thresholded shape-like features may be immune to small noise interference. However, the representational capability of thresholded shape-like features is limited. Only by combining them with traditional texture features is it possible to improve the robustness of DCNNs while not reducing the model's classification accuracy. Therefore, in this paper, we propose a novel neural network model architecture that combines thresholded

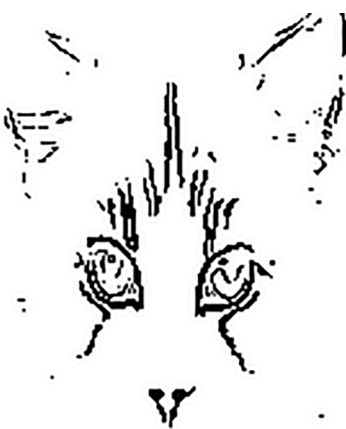

**Fig 3. Thresholded edge image of Fig 1.**

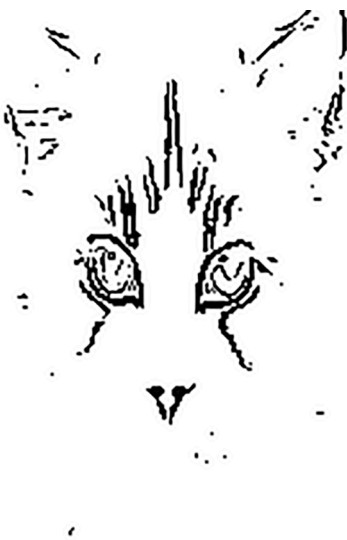

**Fig 4. Thresholded edge image of Fig 2.**

shape-like features and texture features for image classification, aiming to enhance model robustness. The traditional edge detectors, e.g., Sobel [9], DoG [10], Marr-Hildreth [11], and Canny [12], are unlearnable. Considering the learnability of shape-like features, in this paper, four learnable Sobel-like edge detectors are designed and taken as layer kernels, which can be used to extract horizontal, vertical, positive diagonal, and negative diagonal edge features, respectively [13,14]. Note that, in [15], the definition of four difference convolution kernels are similar to the learnable Sobel-like edge detectors in this paper. They both can enhance the details learned by DCNNs. The former is employed for dehazing images, and the latter is leveraged for improving robustness of DCNNs. Based on the four learnable edge detectors, a binary edge feature branch (BEFB) is proposed, which is staked by multiple Sobel layers and one threshold layer. The Sobel layers employ the edge detectors as kernels, and the threshold layer turns the output of the last Sobel layer to the binary features. The binary features concatenated with the texture features learned by any popular backbone are then fed into fully connected layers for classification. To deal with the zero gradient of threshold layer, which makes the weights in BEFB unable to update using the chain rule, Straight-Through Estimator (STE) [16–18] is employed. In addition, to take obfuscated gradient effect raised by [19–21] into consideration, zero gradient or one center-translated sigmoid activation function are also used for generating AEs. In the experiments, we integrate BEFB into VGG16 [22] and ResNet34 [23,24], respectively, and conduct experiments on multiple datasets. The results demonstrate BEFB has no side effects on model training, and BEFB-integrated models can achieve better accuracy under white-box attacks and black-box attack compared to the original ones. Furthermore, we combine the BEFB-integrated models with state-of-the-art (SOTA) robustness enhancement techniques, and find BEFB-integrated models can achieve better accuracy than the original models as well.

The contributions of this paper can be summarized as follows:

1. Inspired by the principal way that the human beings recognize objects, we design four learnable edge detectors and propose BEFB to make DCNNs more robust.
2. BEFB can be easily integrated into any popular backbone, and has no side effects on model training. Extensive experiments on multiple datasets show BEFB-integrated models can achieve better accuracy under white-box attacks and black-box attack compared to the original models.
3. For the first time, we show it is feasible to make DCNNs more robust by combining the shape-like features and texture features.

The organization of the paper is as follows. Section Related Work briefly reviews the related works, and section Proposed Approach describes the details of BEFB. In section Experiments and Discussions, the experiments on multiple datasets are conducted, and the discussions are made. Finally, in section Conclusions, the concluding remarks are given.

## Related work

**Robustness Enhancement with AT.**   AT improves the robustness of the DCNNs by generating AEs as training samples during optimization. Tsipras *et al.* [25] found there exists a tradeoff between robustness and standard accuracy of the models generated by AT, due to the robust classifiers learning a different feature representation compared to the clean classifiers. Madry *et al.* [26] proposed an approximated solution framework to the optimization problems of AT, and found PGD-based AT can produce models defending themselves against the first-order adversaries. Kannan *et al.* [27] investigated the effectiveness of AT at the scale

of ImageNet [28], and proposed a logit pairing AT training method to tackle the tradeoff between robust accuracy and clean accuracy. Wong *et al.* [29] accelerated the training process using FGSM attack with random initialization instead of PGD attack [30], and reached significantly lower cost. Xu *et al.* [31] proposed a novel attack method which can make a stronger perturbation to the input images, resulting in the robustness of models by AT using this attack method is improved. Li *et al.* [32] revealed a link between fast growing gradient of examples and catastrophic overfitting during robust training, and proposed a subspace AT method to mitigate the overfitting and increase the robustness. Dabouei *et al.* [33] found the gradient norm in AT is higher than natural training, which hinders the training convergence of outer optimization of AT. And they proposed a gradient regularization method to improve the performance of AT. Zhou *et al.* [34] proposed an AT framework which can provide detailed and quantifiable explanations through objects, parts, scenes, materials, textures, and colors dimensions. Eleftheriadis *et al.* [35] proposed a general framework for AT to simultaneously consider several distance metrics and adversarial attack strategies.

**Robustness Enhancement without AT.** Non-AT robustness enhancement techniques can be categorized into part-based models [36,37], feature vector clustering [38,39], adversarial margin maximization [40,41], etc. Li *et al.* [36] argued one reason that DCNNs are vulnerable to attacks is they are trained only on category labels, not on the part-based knowledge as humans do. They proposed an object recognition model, which first segments the parts of objects, scores the segmentations based on human knowledge, and final outputs the classification results based on the scores. This part-based model shows better robustness than classic recognition models across various attack settings. Sitawarin *et al.* [37] also thought richer annotation information can help learn more robust features. They proposed a part segmentation model with a head classifier trained end-to-end. The model first segments objects into parts, and then makes predictions based on the parts. Mustafa *et al.* [38] stated that making feature vectors in the same class closer and centroids of different classes more separable can enhance the robustness of DCNNs. They added PCL to the conventional loss function, and designed an auxiliary function to reduce the dimensions of convolutional feature maps. Seo *et al.* [39] proposed a training methodology that enhances the robustness of DCNNs through a constraint that applies a class-specific differentiation to the feature space. The training methodology results in feature representations with a small intra-class variance and large inter-class variances, and can improve the adversarial robustness notably. Yan *et al.* [40] proposed an adversarial margin maximization method to improve DCNNs' generalization ability. They employed the deepfool attack method [42] to compute the distance between an image sample and its decision boundary. This learning-based regularization can enhance the robustness of DCNNs as well. Yuan *et al.* [43] proposed using alpha noise as an alternative to Gaussian noise for data augmentation to improve the robustness of DCNNs. Amerehi *et al.* [44] proposed a label augmentation method to enhance the robustness of DCNNs against both common and intentional perturbations. Dong *et al.* [45] explored the relationship among adversarial robustness, Lipschitz constant, and architecture parameters, and found an appropriate constraint on architecture parameters could reduce the Lipschitz constant to improve the robustness of DCNNs.

Note that, BEFB-integrated models proposed in this paper can be easily combined with above-mentioned robustness enhancement techniques, such as adversarial training (AT) [26] and Prototype Conformity Loss (PCL)[38], and can achieve better classification accuracy than the original models.

## Proposed approach

In this section, we first introduce four learnable edge detectors, and then illustrate a binary edge feature branch (BEFB).

### Learnable edge detectors

Inspired by the observation that humans primarily rely on shape features to recognize objects, here, we design four learnable edge detectors which can extract horizontal edge features, vertical edge features, positive diagonal edge features, and negative diagonal edge features, respectively. These four learnable edge detectors can be taken as layer kernels and are shown in Figs 5–8.

For horizontal edge detector in Fig 5, we have

$$w_{ij} \begin{cases} \in [0, 1] & \text{if } i = 1, j = 1, 2, 3, \\ = 0 & \text{if } i = 2, j = 1, 2, 3, \\ \in [-1, 0] & \text{if } i = 3, j = 1, 2, 3, \end{cases} \tag{1}$$

| $w_{11}$ | $w_{12}$ | $w_{13}$ |
|---|---|---|
| $w_{21}=0$ | $w_{22}=0$ | $w_{23}=0$ |
| $w_{31}$ | $w_{32}$ | $w_{33}$ |

**Fig 5. Horizontal edge detector.**

| $w_{11}$ | $w_{12}=0$ | $w_{13}$ |
|---|---|---|
| $w_{21}$ | $w_{22}=0$ | $w_{23}$ |
| $w_{31}$ | $w_{32}=0$ | $w_{33}$ |

**Fig 6. Vertical edge detector.**

| $w_{11}$ | $w_{12}$ | $w_{13}=0$ |
|---|---|---|
| $w_{21}$ | $w_{22}=0$ | $w_{23}$ |
| $w_{31}=0$ | $w_{32}$ | $w_{33}$ |

**Fig 7. Positive diagonal edge detector.**

| $w_{11}=0$ | $w_{12}$ | $w_{13}$ |
|---|---|---|
| $w_{21}$ | $w_{22}=0$ | $w_{23}$ |
| $w_{31}$ | $w_{32}$ | $w_{33}=0$ |

**Fig 8. Negative diagonal edge detector.**

For vertical edge detector in Fig 6, we have

$$w_{ij} \begin{cases} \in [0, 1] & \text{if } j = 1, i = 1, 2, 3, \\ = 0 & \text{if } j = 2, i = 1, 2, 3, \\ \in [-1, 0] & \text{if } j = 3, i = 1, 2, 3, \end{cases} \quad (2)$$

For positive diagonal edge detector in Fig 7, we have

$$w_{ij} \begin{cases} \in [0, 1] & \text{if } (i, j) \in \{(1, 1), (1, 2), (2, 1)\}, \\ = 0 & \text{if } (i, j) \in \{(1, 3), (2, 2), (3, 1)\}, \\ \in [-1, 0] & \text{if } (i, j) \in \{(2, 3), (3, 2), (3, 3)\}, \end{cases} \quad (3)$$

For negative diagonal edge detector in Fig 8, we have

$$w_{ij} \begin{cases} \in [0, 1] & \text{if } (i, j) \in \{(1, 2), (1, 3), (2, 3)\}, \\ = 0 & \text{if } (i, j) \in \{(1, 1), (2, 2), (3, 3)\}, \\ \in [-1, 0] & \text{if } (i, j) \in \{(2, 1), (3, 1), (3, 2)\}, \end{cases} \quad (4)$$

## Binary edge feature branch

Based on the four learnable edge detectors, BEFB is proposed to extract the binary edge features of the images. BEFB is stacked by multiple Sobel layers and one threshold layer, and can be integrated into any popular backbone. The architecture of a BEFB-integrated DCNN model is shown in Fig 9.

**Sobel Layer.** A Sobel layer is four parallel convolutional layers using addition for fusion. The four parallel convolutional layers take horizontal, vertical, positive diagonal, and negative diagonal edge detectors as their kernels, respectively. Fig 9 shows a BEFB-integrated model in which Sobel layer is with four parallel convolutional layers.

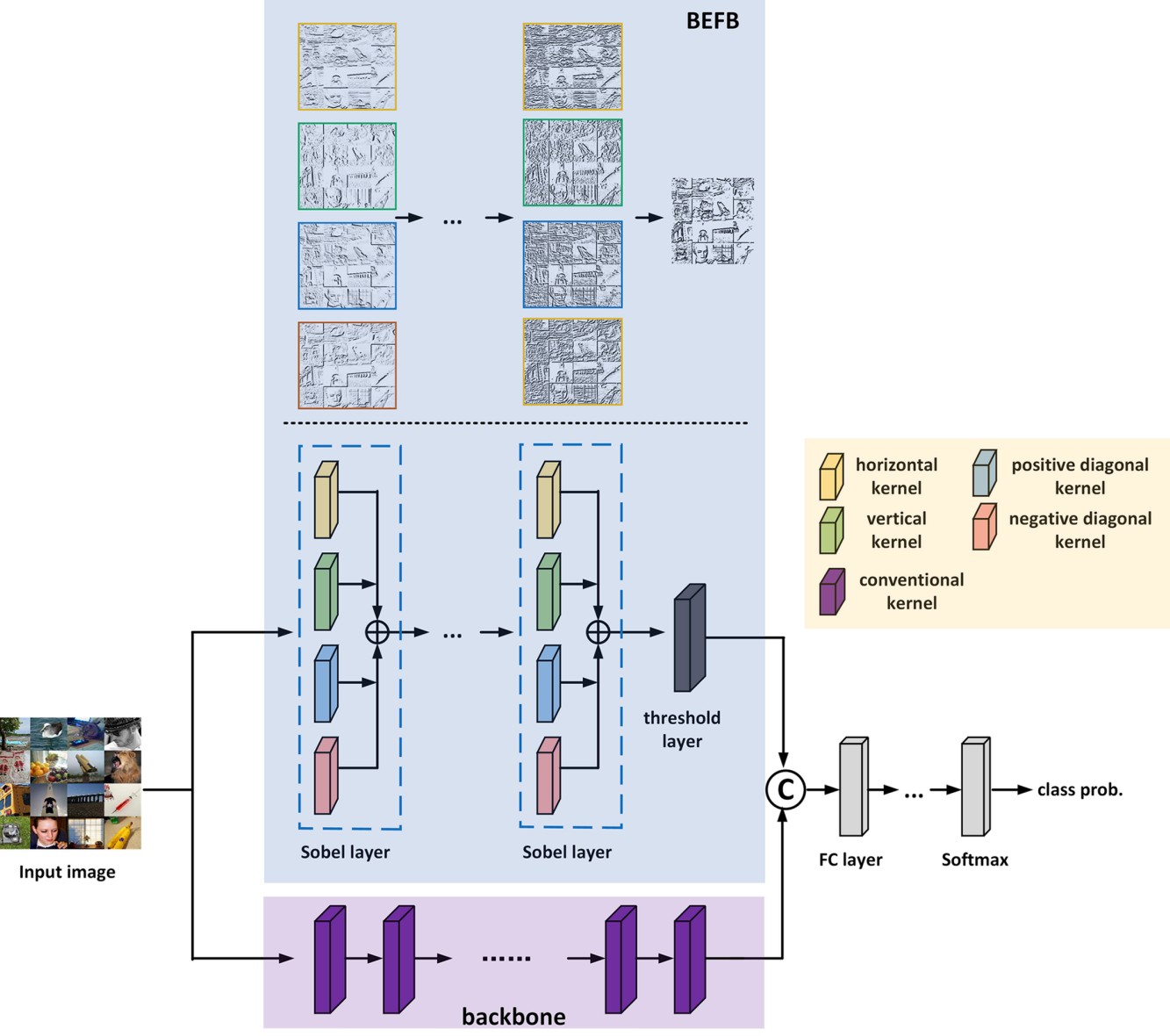

**Fig 9. The architecture of a BEFB-integrated DCNN model.**

**Threshold Layer.** The threshold layer in its nature is an activation layer. The activation function can be written as follows,

$$x_{ikj}^{out} = \begin{cases} 1 & \text{if } x_{ikj}^{in} \geq t * \max(\mathbf{x}_i^{in}), \\ 0 & \text{if } x_{ikj}^{in} < t * \max(\mathbf{x}_i^{in}), \end{cases} \tag{5}$$

In Eq (5), $x_{ikj}^{in}$ is an element of the tensor $\mathbf{x}^{in} \in R^{N \times P \times Q}$, which represents the feature map obtained by the last Sobel layer. $x_{ikj}^{out}$ is an element of the tensor $\mathbf{x}^{out} \in R^{N \times P \times Q}$, which represents the binary feature map output by the threshold layer. $N$ is the number of channels of the feature map, and $P$ and $Q$ are the width and height of a channel. $i = 1, 2, ..., N$, $k = 1, 2, ..., P$, and $j = 1, 2, ..., Q$. $t$ is a proportional coefficient and belongs to $[0, 1]$. $\max(\mathbf{x}_i^{in})$ represents the maximum value of channel $i$. It is obvious to see that, the higher $t$, the less binary features obtained; the smaller $t$, the more binary features obtained.

The threshold layer turns the output of the last Sobel layer to the binary edge features, which concatenated with the texture features learned by the backbone are fed into fully connected layers for classification. Note that the activation function in Eq (5) has a zero gradient. Therefore, the STE technique [16–18] is used to update the weights in BEFB.

## Experiments

In this section, we integrate BEFB into VGG16 [22] and ResNet34 [23,24], respectively, and conduct the experiments on CIFAR-10 [46], MNIST [47], SVHN [48], and TinyImageNet (TinyIN for short) [49] datasets. We examine the effects of BEFB on model training, analyze obfuscated gradient effect [19–21] by using different activation functions of the threshold layer to generate AEs, and compare the accuracy of BEFB-integrated models when facing white-box attacks (FGSM [1], PGD [30], AA[50], $A^3$ [51], and C&W [52]) and black-box attack (One Pixel Attack [53]) with original ones. Furthermore, we combine the BEFB-integrated models with SOTA robustness enhancement techniques to evaluate the classification accuracy. All experiments are coded in Tensorflow with one TITAN XP GPU.

### Experimental settings

In the experiments, the settings of the number of Sobel layers $l$ and proportional coefficient $t$ of threshold layer are shown in Table 1. The settings of the $\epsilon$, *steps*, and *stepsize* of FGSM and PGD are shown in Table 2. In AA and $A^3$, the perturbation limit is set to 20 in MNIST dataset, and 2 in other three datasets. In C&W attack, the constant $c$ is set to 1 for MNIST dataset, and 1e-3 for three other datasets. In black-box attack, three pixels are up to modifications. Each model is run for five times, and the average value is recorded. The number of training epochs is set to 50000. The optimizer used in the experiments is the SGD optimizer, with a momentum of 0.9 and a weight decay coefficient of 5e-4. The initial learning rate is set to 0.01, and the batch size for training samples is set to 128.

### Effects of BEFB on model training

We examine the effects of BEFB on model training by comparing three metrics between the BEFB-integrated models and the original ones, i.e., training accuracy, test accuracy, and training time per epoch. The loss function, optimizer, batch size, and number of epochs are set to be the same. Table 3 shows the comparison of training performance between BEFB-integrated models and the original ones. Table 4 shows the comparison of standard deviation of test accuracy between BEFB-integrated models and the original ones. From both tables, it is clear

**Table 1. The settings of the number of Sobel layers and proportional coefficient of Threshold layer.**

| | CIFAR-10 | | MNIST | | SVHN | | TinyIN | |
|---|---|---|---|---|---|---|---|---|
| | *l* | *t* | *l* | *t* | *l* | *t* | *l* | *t* |
| VGG16-BEFB | 2 | 0.8 | 2 | 0.8 | 2 | 0.8 | 3 | 0.6 |
| ResNet34-BEFB | 2 | 0.6 | 2 | 0.6 | 2 | 0.6 | 3 | 0.6 |

**Table 2. Parameters of FGSM and PGD.**

| | CIFAR-10 | MNIST | SVHN | TinyIN |
|---|---|---|---|---|
| $\epsilon$ | 8 | 80 | 8 | 8 |
| *steps* | 8 | 8 | 8 | 8 |
| *stepsize* | 2 | 20 | 2 | 2 |

**Table 3. Comparison of training performance between BEFB-integrated models and the original ones.**

| | CIFAR-10 | | | MNIST | | | SVHN | | | TinyIN | | |
|---|---|---|---|---|---|---|---|---|---|---|---|---|
| | Tr.Acc. | Te.Acc. | Ti.PE | Tr.Acc. | Te.Acc. | Ti.PE | Tr.Acc. | Te.Acc. | Ti.PE | Tr.Acc. | Te.Acc. | Ti.PE |
| VGG16 | 99.19% | 83.56% | 18s | 99.65% | 99.56% | 17s | 99.39% | 94.46% | 18s | 94.42% | 33.76% | 83s |
| VGG16-BEFB | 99.65% | 83.08% | 18s | 99.66% | 99.56% | 17s | 99.53% | 94.74% | 18s | 92.12% | 30.16% | 93s |
| ResNet34 | 99.23% | 79.23% | 47s | 99.73% | 99.51% | 42s | 99.70% | 93.18% | 51s | 97.48% | 30.75% | 243s |
| ResNet34-BEFB | 99.51% | 74.65% | 46s | 99.45% | 99.36% | 41s | 99.81% | 93.38% | 50s | 98.01% | 26.34% | 248s |

**Tr.Acc.** stands for training accuracy. **Te.Acc.** stands for test accuracy. **Ti.PE** stands for training time per epoch.

**Table 4. Comparison of standard deviation of test accuracy between BEFB-integrated models and original ones.**

| | CIFAR-10 | MNIST | SVHN | TinyIN |
|---|---|---|---|---|
| VGG16 | 0.0011 | 0.0007 | 0.0010 | 0.0012 |
| VGG16-BEFB | 0.0009 | 0.0008 | 0.00011 | 0.0010 |
| ResNet34 | 0.0013 | 0.0011 | 0.0012 | 0.0014 |
| ResNet34-BEFB | 0.0010 | 0.0009 | 0.0013 | 0.0015 |

to see the BEFB-integrated models perform comparably with the original models, which indicates BEFB is lightweight and has no side effects on model training. Figs 10 and 11 compare the training profiles of VGG16 and VGG16-BEFB on CIFAR-10 dataset, respectively. Figs 12 and 13 compare the training profiles of ResNet34 and ResNet34-BEFB on CIFAR-10 dataset, respectively. From the figures, we can see BEFB-integrated models demonstrate the similar training dynamics with the original ones. Note that, the training epochs for all four models in Figs 10–13 is 50000. To more clearly show the comparison of converging speed and accuracy between BEFB-integrated models and original models, Figs 10–13 illustrate the changes in training loss and accuracy for the first 60 epochs.

## Analysis of obfuscated gradient effect

The activation function of the threshold layer is expressed by Eq (5), whose gradient is zero. In order to update the weights in BEFB, STE technique [16–18] is adopted. But with respect to generating AEs, STE may bring obfuscated gradient effect [19–21], which means the yielded AEs are not powerful enough to deceive the models. Here, in addition to STE, zero gradient and gradient of a center-translated sigmoid activation function for threshold layer are also

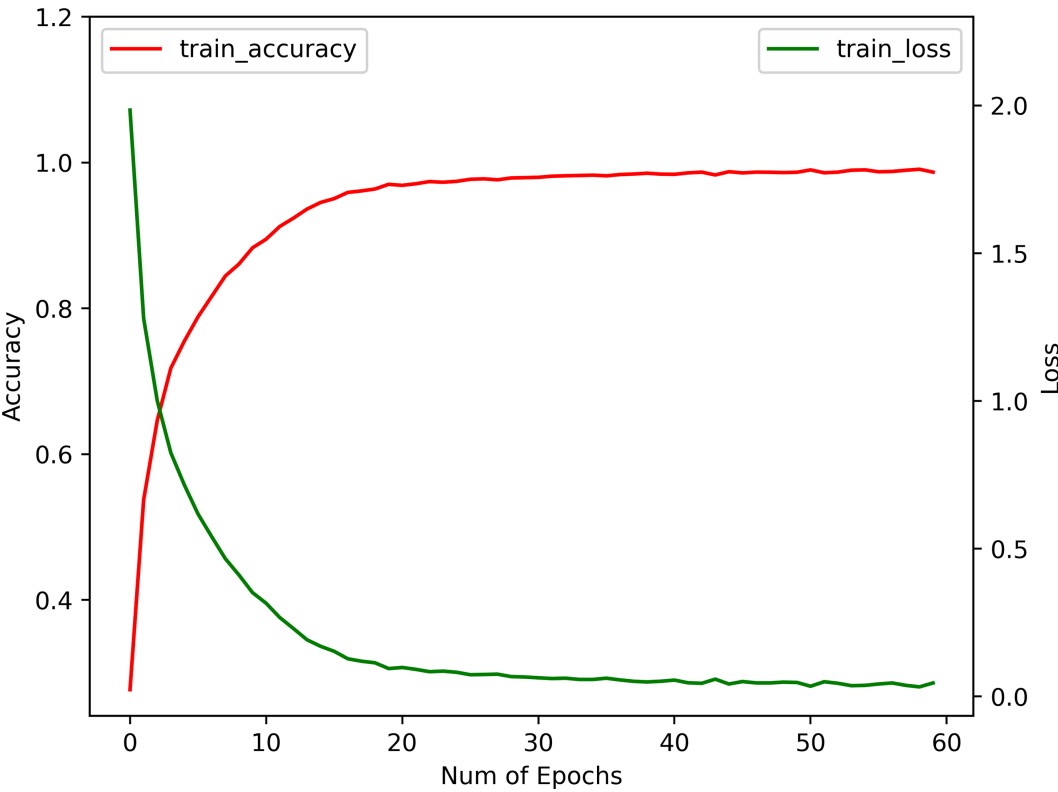

**Fig 10. Training profile of VGG16.**

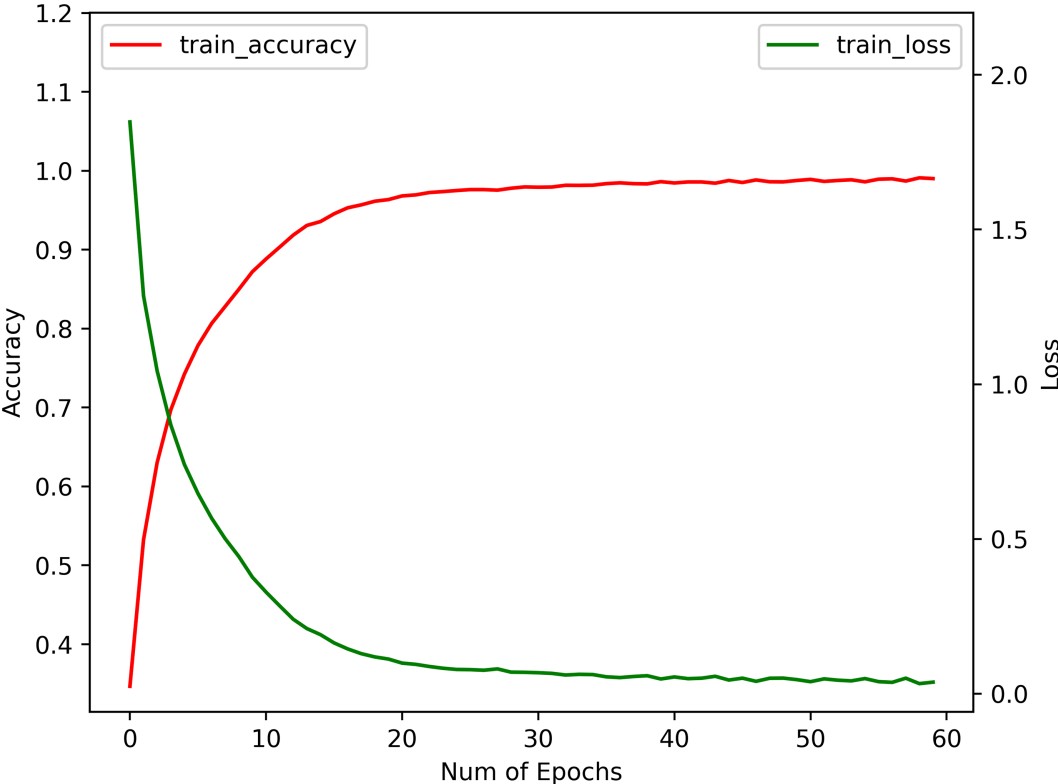

**Fig 11. Training profile of VGG16-BEFB.**

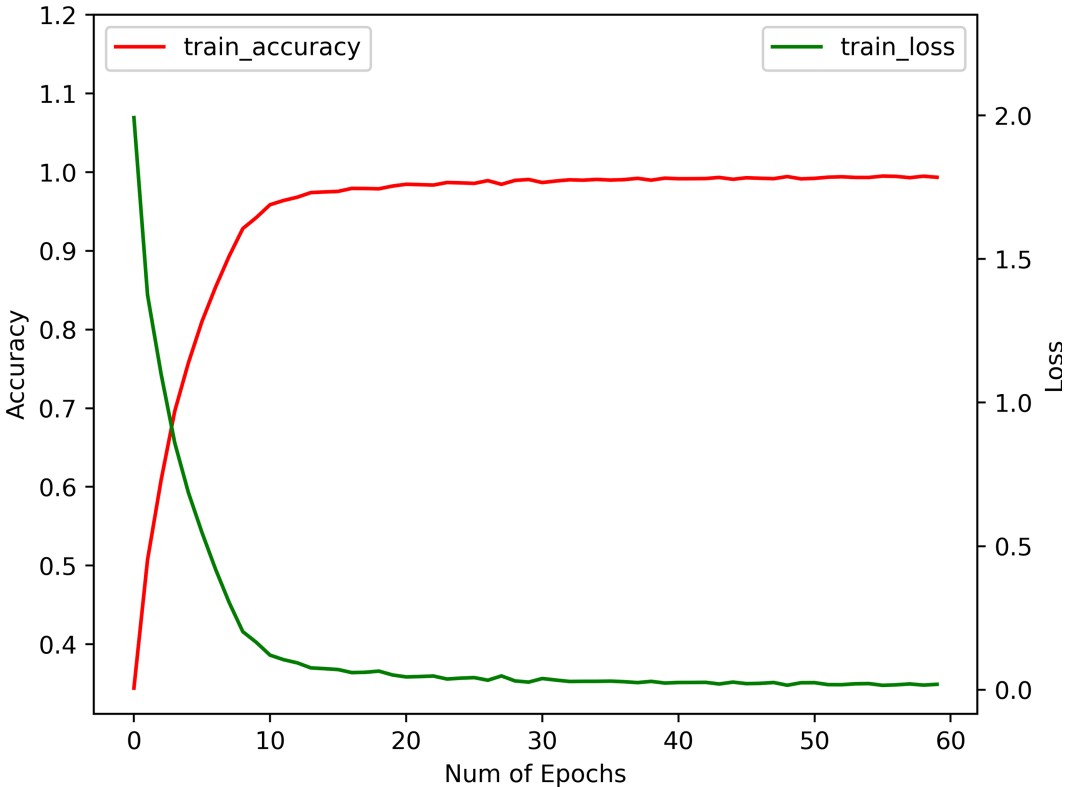

**Fig 12. Training profile of ResNet34.**

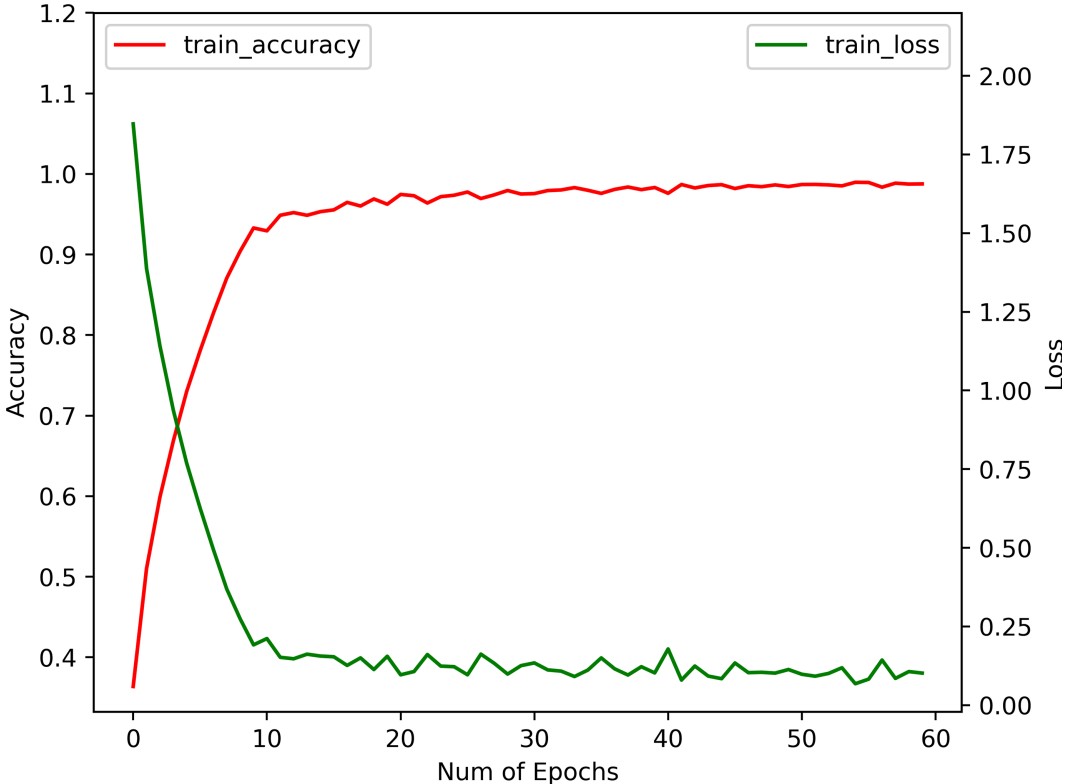

**Fig 13. Training profile of ResNet34-BEFB.**

tested. Table 5 compares the classification accuracy of AEs generated by these three gradient computing strategies under FGSM attacks. Eq (6) defines the center-translated sigmoid function. $x_0$ in general, is the threshold value for channels, illustrated in Eq (5).

$$f(x) = \frac{1}{1 + e^{x - x_0}} \tag{6}$$

From the table 5, it is clear to see the classification accuracy under STE is much higher than zero gradient and gradient of center-translated sigmoid function. It indeed brings the obfuscated gradient effect using STE. In our experiments, we employ zero gradient of threshold layer to generate AEs.

## Performance comparison between BEFB-integrated models and original models under attacks

Tables 6 and 7 compare the classification accuracy of BEFB-integrated models and the original models under white-box attacks, in which FGSM [1], PGD [30], AA [50], $A^3$ [51], and C&W attacks are employed. Table 8 compares the classification accuracy of BEFB-integrated models and the original models under black-box attack [53].

From the tables, it is clear to see that BEFB-integrated models are more robust than the original models under white-box attacks and black-box attack. For white-box attacks, on CIFAR-10 dataset, VGG16-BEFB model can achieve 17%, 6%, 7%, and 4% higher accuracy

**Table 5. Comparison of classification accuracy of AEs generated by three gradient computing strategies under FGSM attacks.**

| | | CIFAR-10 | MNIST | SVHN | TinyIN |
|---|---|---|---|---|---|
| VGG16-BEFB | STE | 64.24% | 81.65% | 71.02% | 43.21% |
| | Sigmoid | 44.93% | 68.25% | 59.75% | 33.06% |
| | Zero Gradient | **44.40%** | **67.42%** | **59.28%** | **31.97%** |
| ResNet34-BEFB | STE | 23.35% | 17.55% | 14.36% | 35.87% |
| | Sigmoid | **14.87%** | **13.84%** | 12.71% | 5.82% |
| | Zero Gradient | 14.88% | 14.53% | **12.17%** | **5.39%** |

**Table 6. Comparison of classification accuracy between BEFB-integrated models and original ones under white-box attacks.**

| | CIFAR-10 | | | | MNIST | | | | SVHN | | | | TinyIN | | | |
|---|---|---|---|---|---|---|---|---|---|---|---|---|---|---|---|---|
| | FGSM | PGD | AA | $A^3$ | FGSM | PGD | AA | $A^3$ | FGSM | PGD | AA | $A^3$ | FGSM | PGD | AA | $A^3$ |
| VGG16 | 28.59% | 1.93% | 7.39% | 7.16% | 56.54% | 3.63% | 38.77% | 38.86% | 57.77% | 16.84% | 48.36% | 48.35% | 27.39% | 4.03% | 2.47% | 2.41% |
| VGG16-BEFB | **45.60%** | **8.05%** | **14.57%** | **11.84%** | **67.42%** | **15.53%** | **76.04%** | **76.13%** | **59.28%** | **19.48%** | **53.85%** | **53.77%** | **31.93%** | **9.18%** | **2.85%** | **2.79** |
| ResNet34 | 7.82% | 0.47% | 15.16% | 14.82% | 12.66% | 0.76% | 25.77% | 25.75% | 24.14% | 2.49% | 49.36% | 49.30% | 5.80% | 0.00% | **5.14%** | 5.07% |
| ResNet34-BEFB | **14.88%** | **1.01%** | **15.83%** | **15.62%** | **14.53%** | **2.95%** | **72.43%** | **72.33%** | **25.17%** | **2.98%** | **51.35%** | **50.97%** | **5.95%** | **0.95%** | 5.08% | **5.09%** |

**Table 7. Comparison of classification accuracy between BEFB-integrated models and original ones under C&W attack.**

| | CIFAR-10 | MNIST | SVHN | TinyIN |
|---|---|---|---|---|
| VGG16 | 56.94% | 71.41% | 82.16% | 18.32% |
| VGG16-BEFB | **59.13%** | **74.12%** | **83.01%** | **19.44%** |
| ResNet34 | 57.74% | 68.24% | 81.23% | 21.64% |
| ResNet34-BEFB | **60.49%** | **69.37%** | **81.60%** | **22.30%** |

**Table 8. Comparison of classification accuracy between BEFB-integrated models and original ones under black-box attack.**

|              | CIFAR-10 | MNIST  | SVHN   | TinyIN |
|--------------|----------|--------|--------|--------|
| VGG16        | 16.20%   | 94.40% | 37.80% | 54.40% |
| VGG16-BEFB   | **29.40%** | **95.80%** | **40.60%** | **55.00%** |
| ResNet34     | 40.60%   | 69.20% | 35.60% | 66.60% |
| ResNet34-BEFB | **41.20%** | **78.60%** | **41.00%** | **66.80%** |

than the original model under FGSM, PGD, AA, and $A^3$ attacks, respectively. On MNIST dataset, VGG16-BEFB model can achieve 11%, 12%, 37%, and 37% higher accuracy than the original model under FGSM, PGD, AA, and $A^3$ attacks, respectively. For black-box attack, on CIFAR-10 dataset, VGG16-BEFB model can achieve 13% higher accuracy than the original model. On MNIST dataset, ResNet34-BEFB model can achieve 9% higher accuracy than the original model.

Table 9 integrates BEFB with WRN34-10 [54] backbone, and the results show the classification accuracy of WRN34-10-BEFB is better than the original WRN34-10 under FGSM, PGD, AA, $A^3$ attacks on CIFAR-10 dataset.

Figs 14–17 depict how BEFB helps improve the robustness of VGG16 model. In these figures, the gray features are extracted from backbone branch, and the binary features are extracted from BEFB. Figs 14 and 15 examine the extracted features of both clean examples and the AEs by the original VGG16 model and VGG16-BEFB model on TinyIN dataset, respectively. On Fig 14, the certain images from TinyIN dataset under FGSM of $\epsilon$=16 attack

**Table 9. Comparison of classification accuracy between WRN34-10-BEFB model and original one on CIFAR-10 dataset.**

|              | FGSM   | PGD    | AA     | $A^3$   |
|--------------|--------|--------|--------|--------|
| WRN34-10     | 4.13%  | 0.00%  | 9.25%  | 9.12%  |
| WRN34-10-BEFB | **5.01%** | 0.00%  | **9.42%** | **9.33%** |

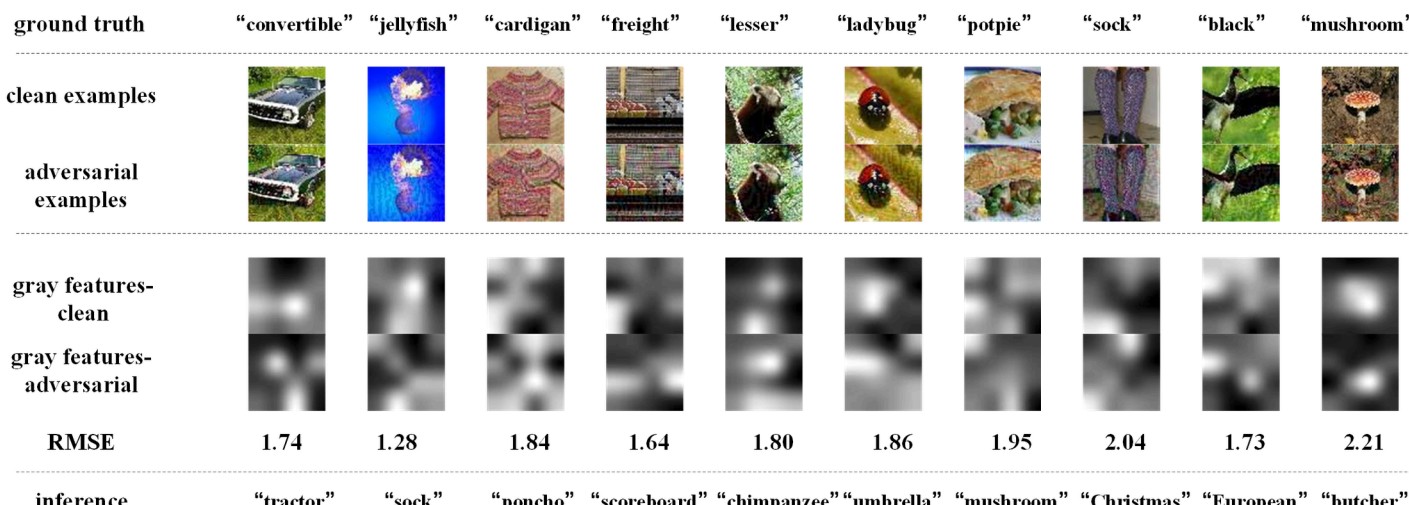

**Fig 14. Texture features extracted by the original VGG16 model and its predictions under FGSM of $\epsilon$=16 attack.**

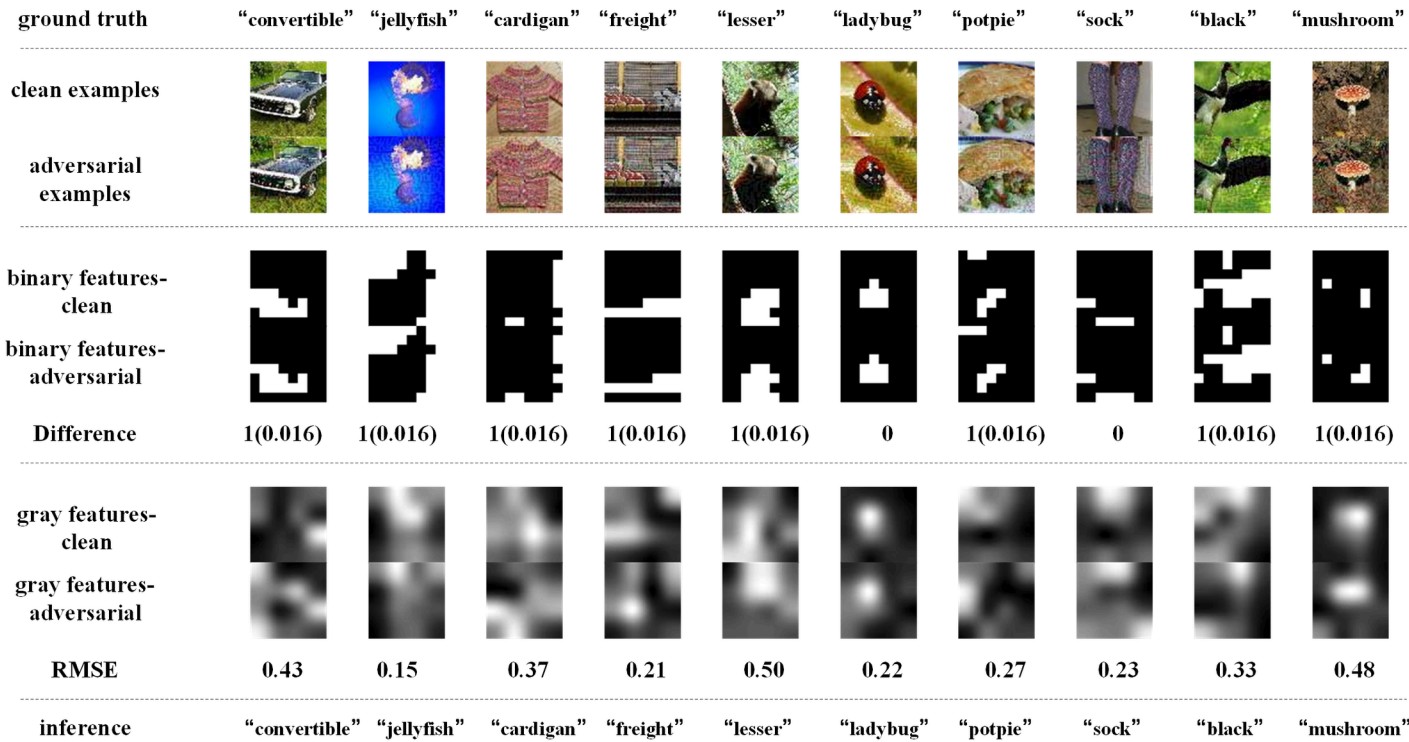

**Fig 15. Binary edge features and the texture features extracted by the VGG16-BEFB model and its predictions under FGSM of $\epsilon$=16 attack.**

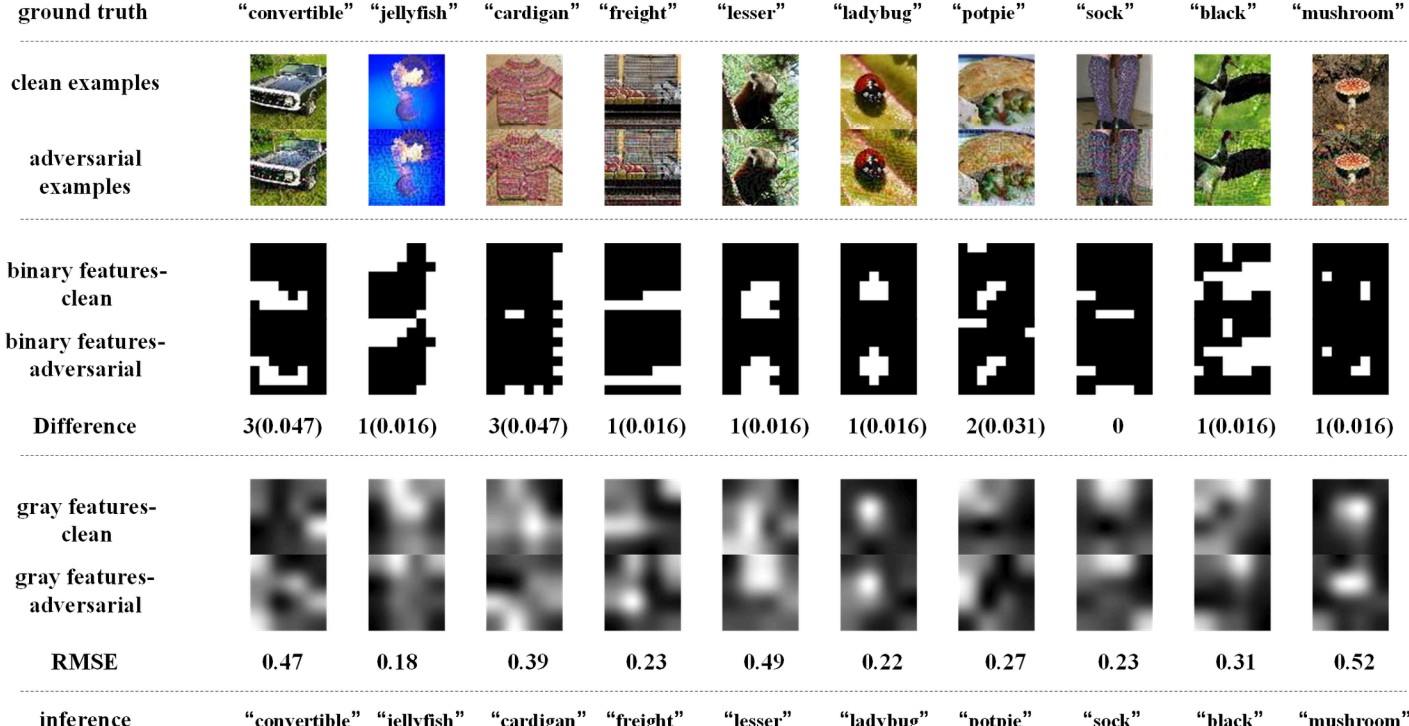

**Fig 16. Binary edge features and the texture features extracted by the VGG16-BEFB model and its predictions under FGSM of $\epsilon$=20 attack.**

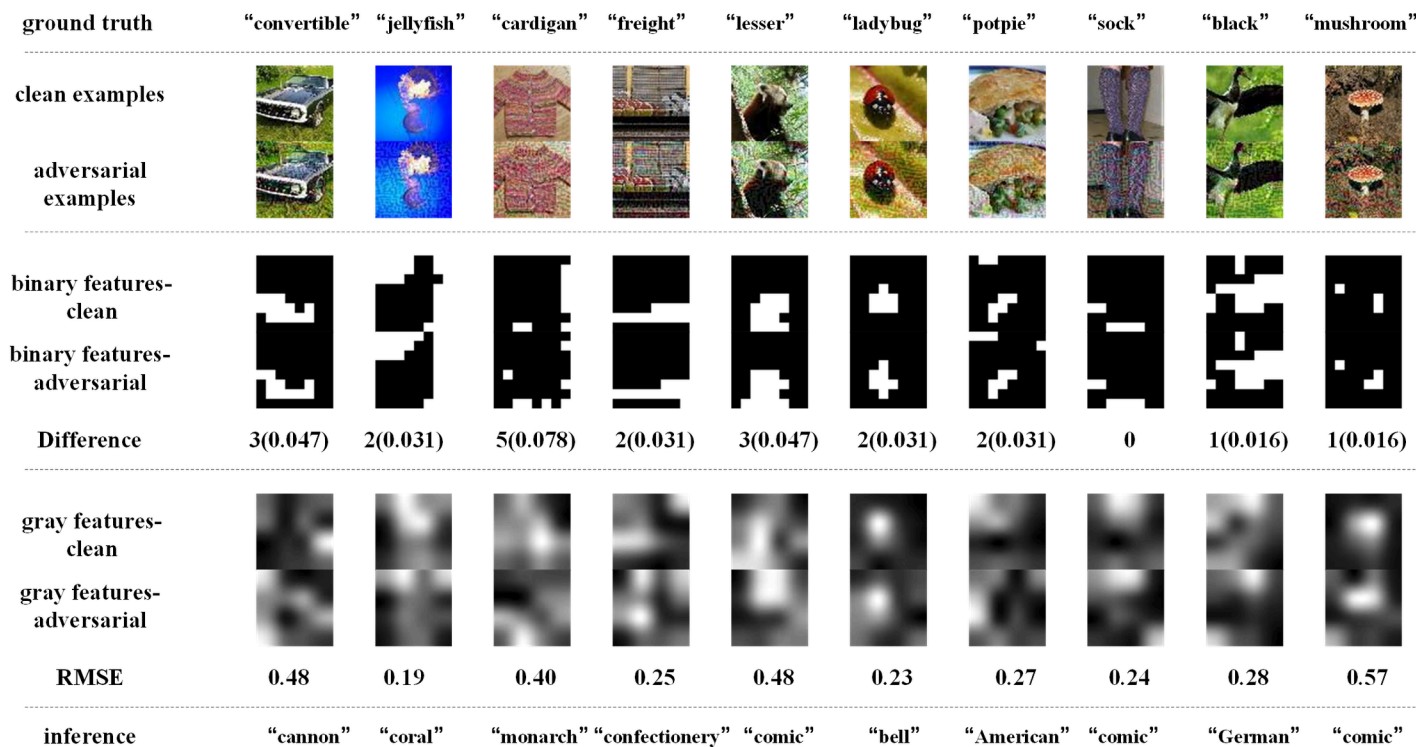

**Fig 17. Binary edge features and the texture features extracted by the VGG16-BEFB model and its predictions under FGSM of $\epsilon$=24 attack.**

are predicted wrong by the original VGG16 model, and it is clear to see there is significant difference between the extracted texture features of the clean examples and the AEs. On Fig 15, the same images from TinyIN dataset under FGSM of $\epsilon$=16 attack are classified correctly by the VGG16-BEFB model, and the difference of the extracted texture features between the clean examples and the AEs is lower than that in the original model. More notably, the extracted binary edge features of clean examples and AEs are almost the same, e.g., in "ladybug" image and "sock" image, the number of different pixels are both zero. It indicates these extracted binary edge features play an important role in improving the classification accuracy on AEs. Furthermore, we add more perturbations to the images and observe the extracted features and prediction results of the VGG16-BEFB models. On Fig 16, four more perturbations are added on the images under FGSM attack. The difference of texture features between the clean examples and AEs is slightly higher than that under $\epsilon$=16 attack, e.g., RMSE (root mean square error) of "jellyfish" image is 0.15 under $\epsilon$=16 and 0.18 under $\epsilon$=20, and RMSE of "mushroom" image is 0.48 under $\epsilon$=16 and 0.52 under $\epsilon$=20. The difference of binary edge features between the clean examples and AEs is nearly unchanged under $\epsilon$=16 attack and $\epsilon$=20 attack, e.g., the number of different pixels for "jellyfish" image is 1 under both $\epsilon$=16 attack and $\epsilon$=20 attack, and the number of different pixels for "mushroom" image is also 1 under both $\epsilon$=16 attack and $\epsilon$=20 attack. And the prediction results under FGSM of $\epsilon$=20 attack are correct. On Fig 17, another four perturbations are added, and the prediction results are wrong. It can be seen from the figure that, the difference of texture features under $\epsilon$=24 attack is slightly higher than that under $\epsilon$=20 attack, e.g., RMSE of "jellyfish" image is 0.18 under $\epsilon$=20 and 0.19 under $\epsilon$=24, and RMSE of "mushroom" image is 0.52 under $\epsilon$=20 attack and 0.57 under $\epsilon$=24 attack. The number of different pixels in binary edge features of clean

examples and AEs keeps very close under $\epsilon=20$ attack and $\epsilon=24$ attack, e.g. the number for "mushroom" image keeps unchanged and the number for "jellyfish" image increases one. Both Figs 16 and 17 demonstrate the binary edge features are less susceptible to small perturbations, and to further improve the classification accuracy on AEs, exploring the novel combination forms for both texture features and binary edge features is a key issue.

Similarly, Figs 18–21 depict how BEFB helps improve the robustness of ResNet34 model. On Figs 18 and 19, it can be seen that, the original ResNet34 model makes the wrong predictions for the certain images from CIFAR-10 dataset under FGSM of $\epsilon=8$ attack, while ResNet34-BEFB model gets them correct because of the extracted binary edge features almost the same between the clean examples and the AEs. For example, on "dog", "airplane", "automobile", "ship" and "frog" images, the number of different pixels is zero. On Figs 20 and 21, four more and eight more perturbations are added, respectively. It can be seen that the prediction results are correct under FGSM of $\epsilon=12$ attack and wrong under FGSM of $\epsilon=16$ attack. When focusing on the binary edge features under these two attacks, it is found the changes on binary edge features in most of cases are zeros, e.g., in "dog", "truck", and "ship" images, demonstrating the binary edge features are less susceptible to small perturbations.

We also compare the classification performance of both original models and BEFB-integrated models on the images perturbed by gaussian noise. Figs 22 and 23 illustrate the gaussian noise perturbed images of CIFAR-10 dataset and MNIST dataset. In Table 10, it is clear to see the classification accuracy of BEFB-integrated models are better than the original models on both datasets, e.g. the ResNet34-BEFB model can achieve 15% and 10% higher accuracy than the original model on CIFAR-10 dataset and MNIST dataset, respectively.

## Combining BEFB-integrated models with SOTA robustness enhancement methods

We combine BEFB-integrated models with four popular robustness enhancement techniques, i.e., AT [26], PCL [38], DDPM [55], and BORT [56], and compare them to the original models with these robustness enhancement techniques. We denote VGG16-BEFB models with AT as VGG16-BEFB-AT, and VGG16-BEFB models with PCL as VGG16-BEFB-PCL. The same

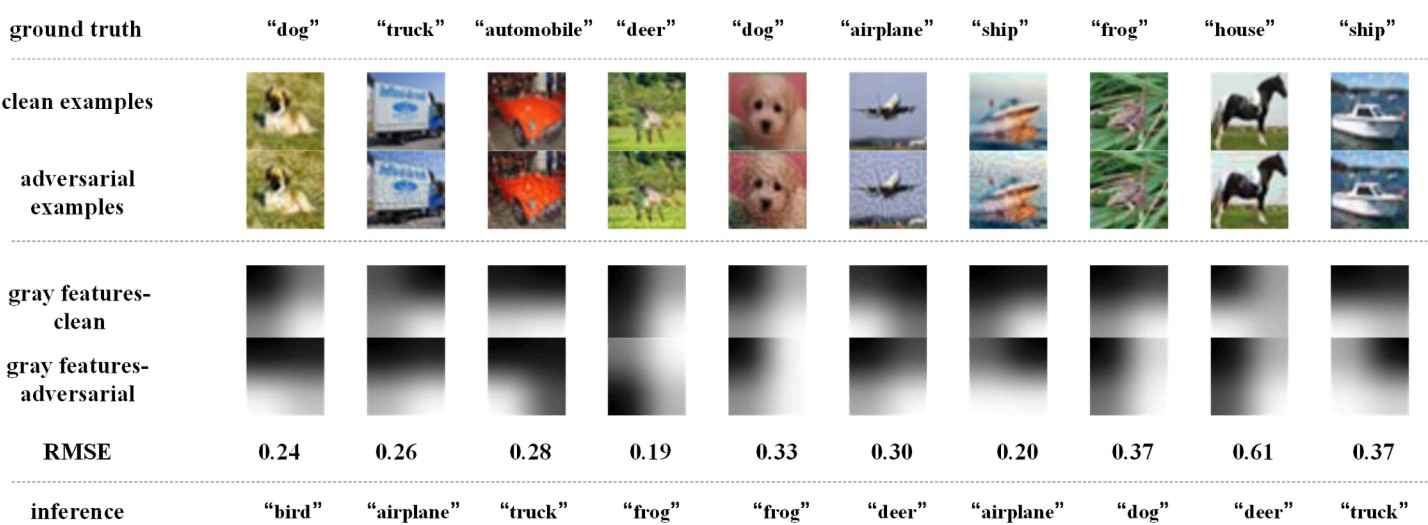

**Fig 18. Texture features extracted by the original ResNet34 model and its predictions under FGSM of $\epsilon=8$ attack.**

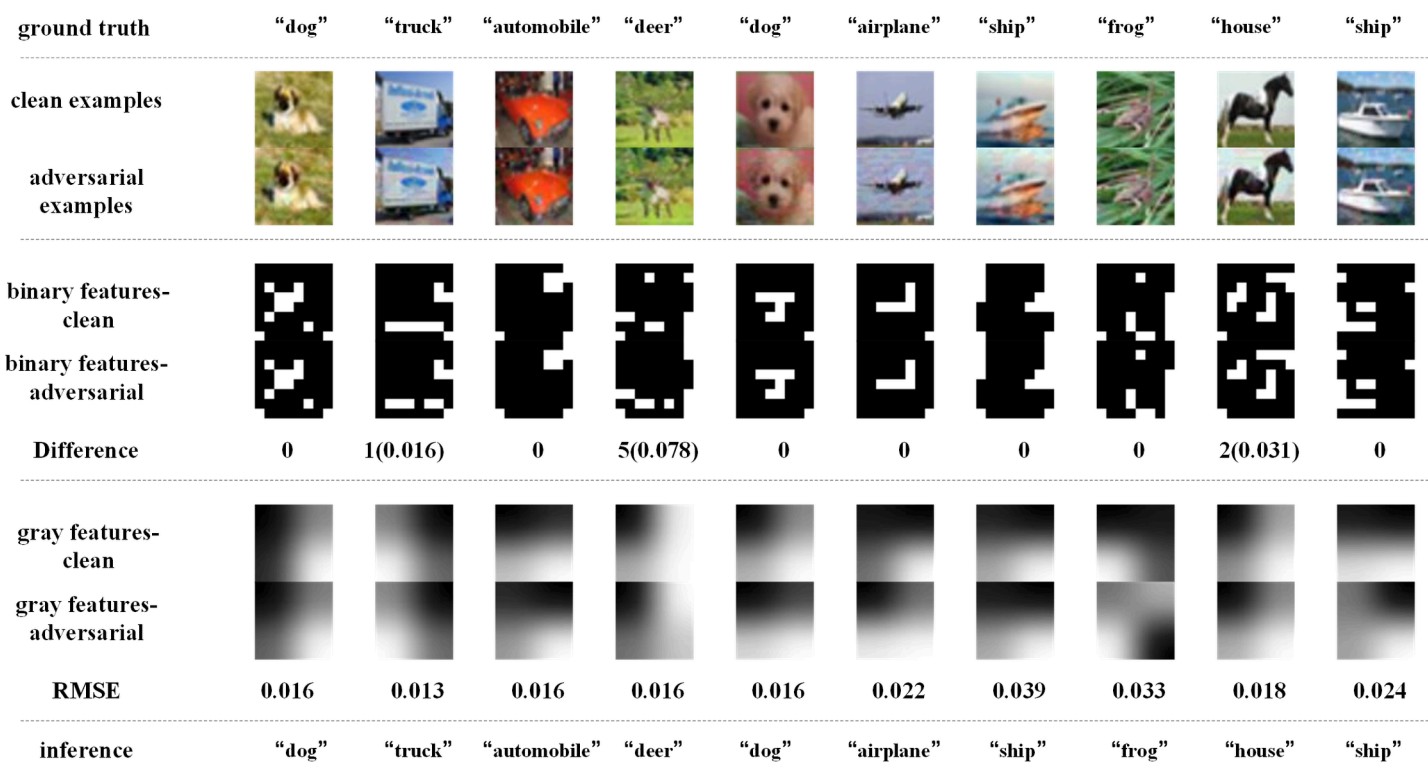

**Fig 19. Binary edge features and the texture features extracted by the ResNet34-BEFB model and its predictions under FGSM of *ε*=8 attack.**

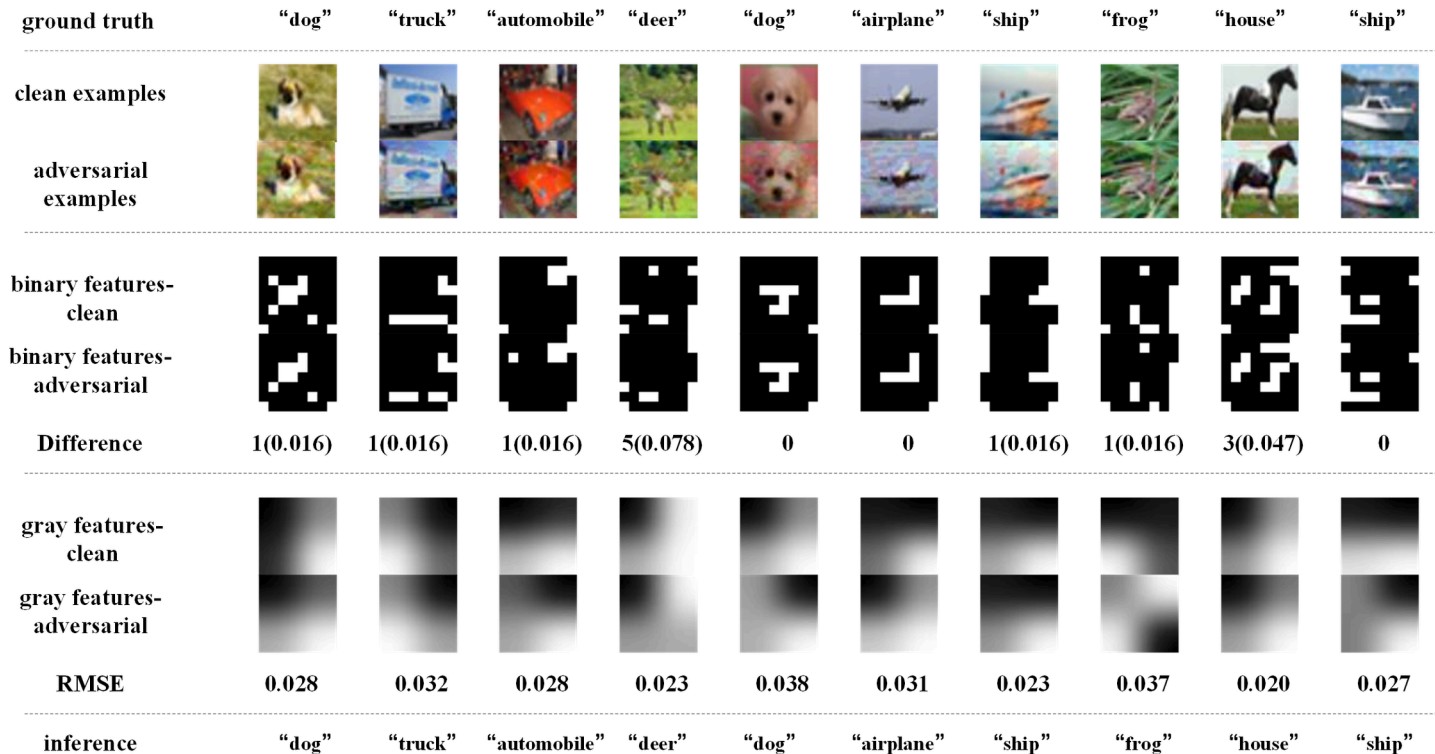

**Fig 20. Binary edge features and the texture features extracted by the ResNet34-BEFB model and its predictions under FGSM of *ε*=12 attack.**

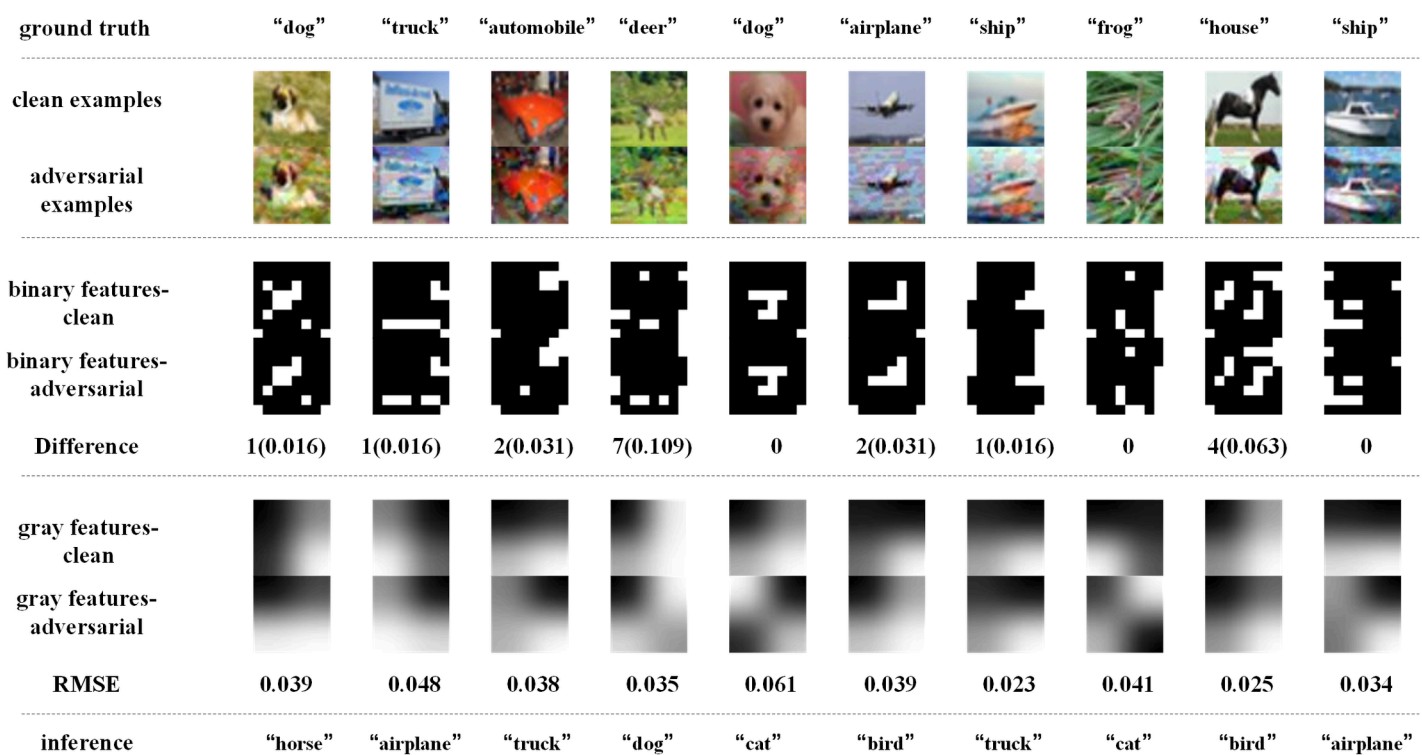

**Fig 21. Binary edge features and the texture features extracted by the ResNet34-BEFB model and its predictions under FGSM of $\varepsilon$=16 attack.**

notations for ResNet34-BEFB models and the original models. Fig 24 compares the classification accuracy of BEFB-AT models with AT enhanced original models under PGD attack. Figs 25 and 26 compare the classification accuracy of BEFB-PCL models with PCL enhanced original models under FGSM and PGD attacks.

From the figures, we can see AT and PCL enhanced BEFB-integrated models have better classification accuracy than AT and PCL enhanced original models.

In Table 11, DDPM and BORT are integrated with WRN28-10 [54] and WRN28-10-BEFB models respectively for robustness enhancement. The results show DDPM/BORT enhanced WRN28-10-BEFB model has higher classification accuracy than original WRN28-10 model under FGSM, PGD, AA, and A$^3$ attacks.

## Ablation study

We first make a comparison between BEFB-integrated models with learnable edge detectors and those with traditional Sobel edge detector. BEFB-integrated models with traditional Sobel edge detector are denoted as BEFB-fixed. Table 12 shows the comparison of classification accuracy between the original BEFB-integrated models and BEFB-integrated models with traditional Sobel edge detector. From the table, it can be seen that the average classification accuracy decreases when replacing the learnable edge detectors with traditional Sobel edge detector.

Considering BEFB-integrated models having two main components, i.e., Sobel layers and threshold layer, we then make a comparison between original BEFB-integrated models and threshold layer and Sobel layer removed models. Threshold layer removed models are denoted as BEFB-tlre, and Sobel layers removed models are denoted as BEFB-slre. Table 13 shows

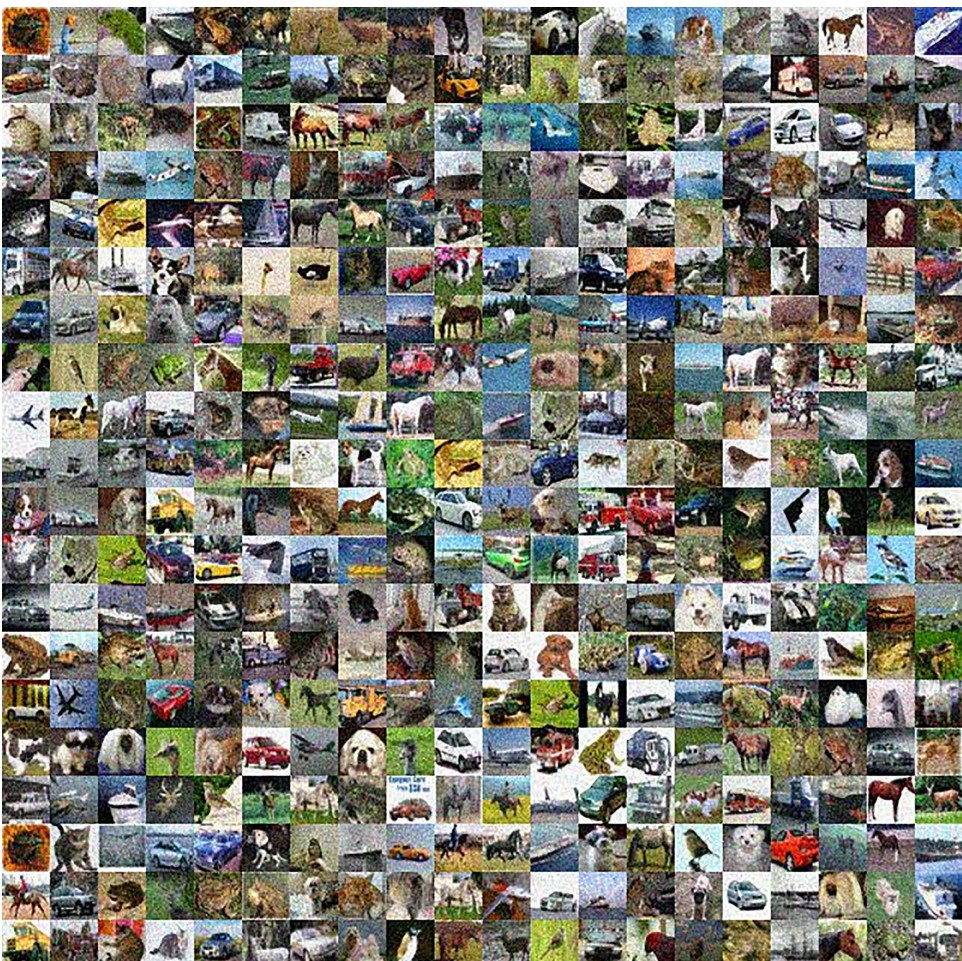

**Fig 22. Gaussian noise with zero mean and 0.08 standard deviation on CIFAR-10 dataset.**

the comparison of classification accuracy between the original BEFB-integrated models and BEFB-integrated models with threshold layer and Sobel layers removed, respectively. From the table, it is clear to see when removing threshold layer and Sobel layers, the robustness of BEFB-tlre and BEFB-slre models is weakened.

## Discussions

The experimental results from Section shows BEFB has no side effects on model training, and BEFB-integrated models are more robust than original models under attacks. And when combining BEFB-integrated models with SOTA robustness enhancement techniques, it can achieve better classification accuracy than original models. It also can be seen that, using BEFB to enhance robustness is effective but not that significant. It may be because under BEFB, the binary edge features are combined with texture features by concatenation. We believe it is worthwhile to explore other combination forms of binary edge features and texture features which can potentially improve the robustness of DCNNs notably.

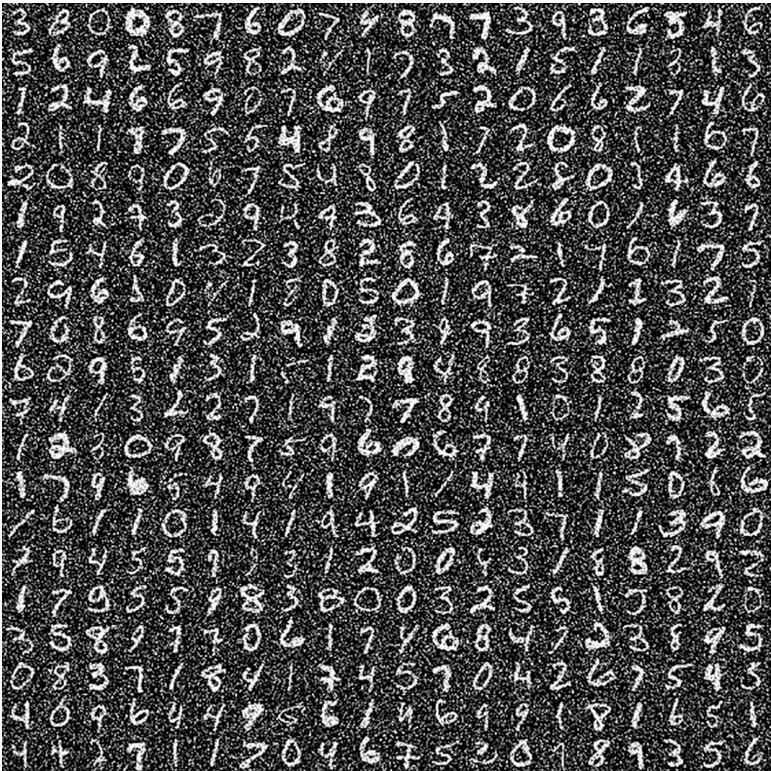

**Fig 23. Gaussian noise with zero mean and 0.35 standard deviation on MNIST dataset.**

**Table 10. Comparison of classification accuracy between the original models and BEFB-integrated models on gaussian noise perturbed CIFAR-10 and MNIST datasets.**

|  | CIFAR-10 | MNIST |
|---|---|---|
| VGG16 | 61.35% | 85.17% |
| VGG16-BEFB | **64.17%** | **88.75%** |
| ResNet34 | 75.63% | 76.01% |
| ResNet34-BEFB | **91.62%** | **86.11%** |

## Conclusions

Enhancing the robustness of DCNNs is of great significance for the safety-critical applications in the real world. Inspired by the principal way that human eyes recognize objects, in this paper, we design four learnable edge detectors and propose a binary edge feature branch (BEFB), which can be easily integrated into any popular backbone. Experiments on multiple datasets show BEFB has no side effects on model training, and BEFB-integrated models are more robust than the original models. The work in this paper for the first time shows it is feasible to combine shape-like features and texture features to make DCNNs more robust. In future's work, we endeavor to explore other effective and efficient combination forms of binary edge features and texture features, and design an optimization framework for the parameter searching to yield models with good performance under attacks.

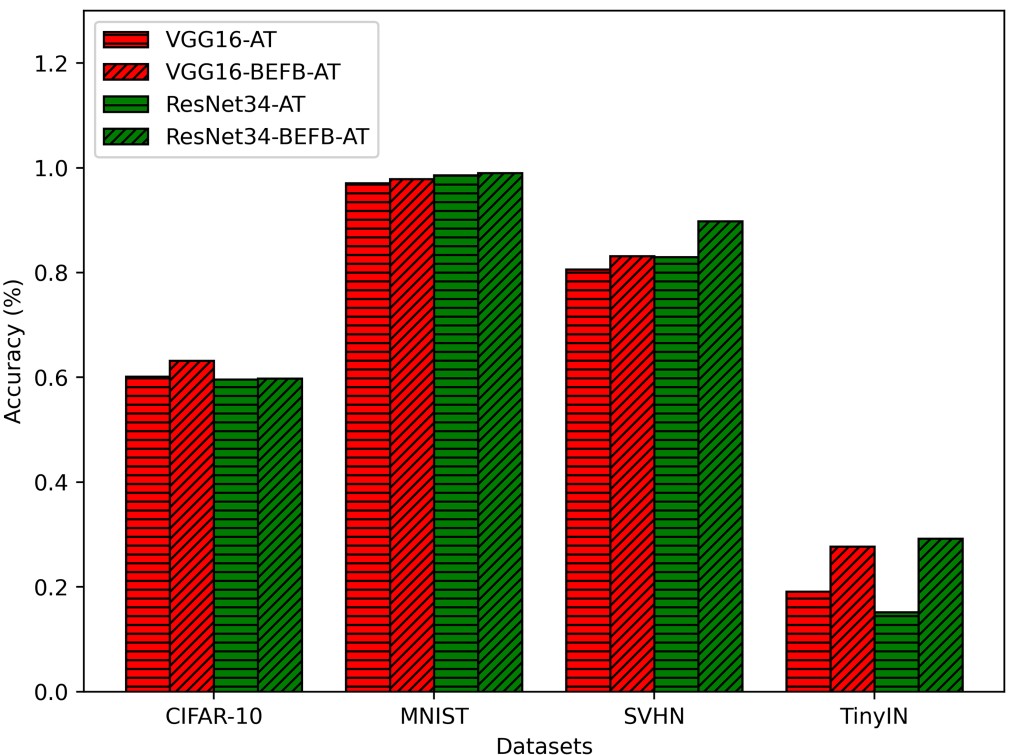

**Fig 24. Comparing AT enhanced BEFB-integrated models with original models under PGD attack.**

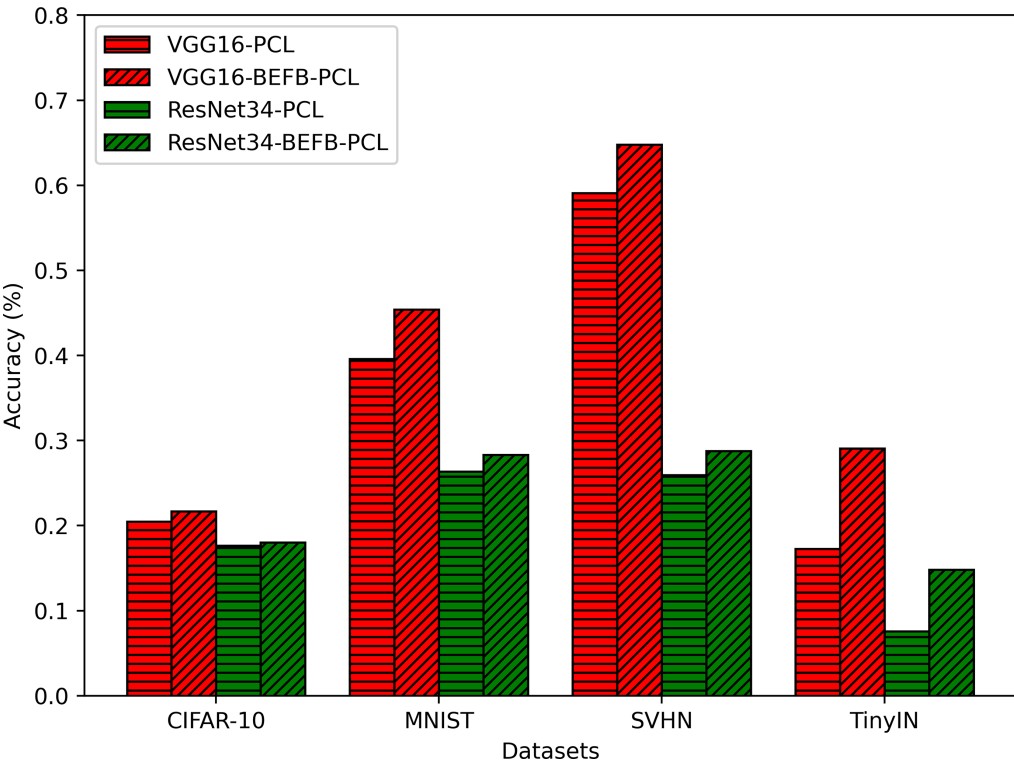

**Fig 25. PCL enhanced BEFB-integrated models and original models under FGSM attack.**

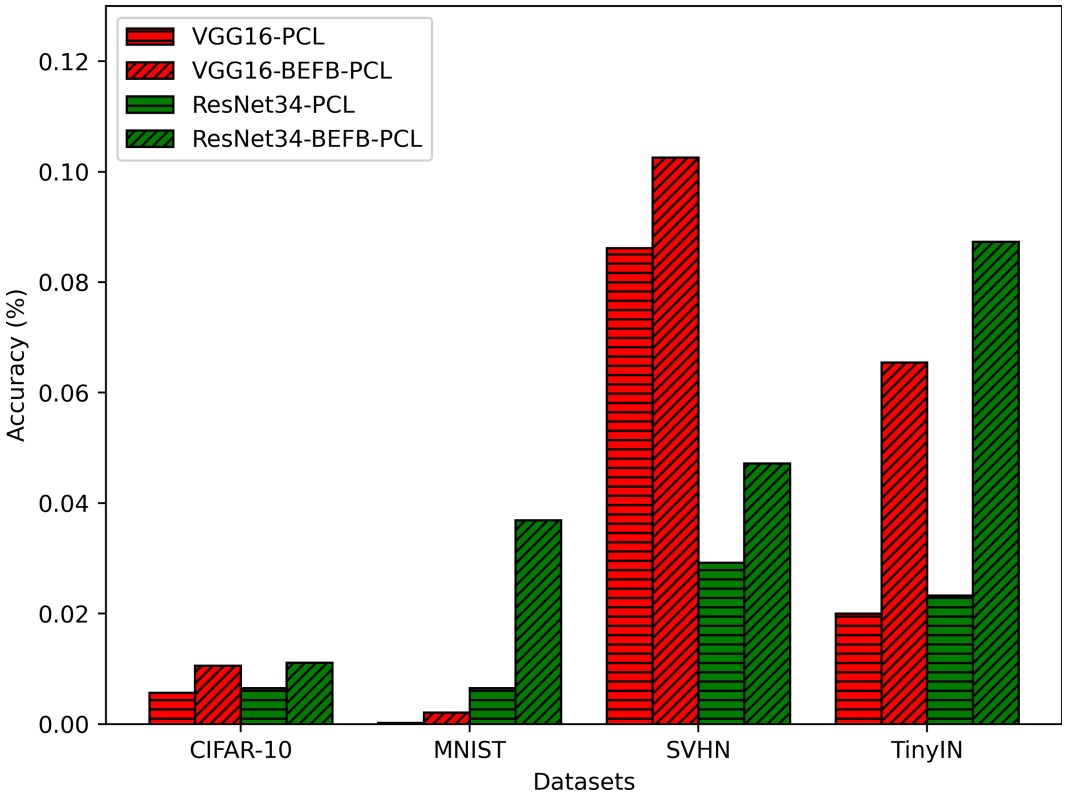

**Fig 26. PCL enhanced BEFB-integrated models and original models under PGD attack.**

**Table 11. Comparison of classification accuracy between DDPM/BORT enhanced WRN28-10-BEFB model and original one under FGSM, PGD, AA, and $A^3$ attacks.**

|  | FGSM | PGD | AA | $A^3$ |
|---|---|---|---|---|
| **WRN28-10-DDPM** | 75.78% | 71.34% | 69.07% | 68.99% |
| **WRN28-10-BEFB-DDPM** | **76.49%** | **71.81%** | **69.55%** | **69.47%** |
| **WRN28-10-BORT** | 58.29% | 52.95% | 50.29% | 50.20% |
| **WRN28-10-BEFB-BORT** | **58.71%** | **53.39%** | **50.54%** | **50.53%** |

**Table 12. Comparison of replacing learnable edge detectors with traditional Sobel edge detector.**

|  | CIFAR-10 | | SVHN | |
|---|---|---|---|---|
|  | FGSM | PGD | FGSM | PGD |
| **VGG16-BEFB-fixed** | 42.90% | 3.98% | **60.72%** | 17.83% |
| **VGG16-BEFB** | **45.60%** | **8.05%** | 59.28% | **19.48%** |
| **ResNet34-BEFB-fixed** | 11.37% | **1.09%** | 23.12% | **3.21%** |
| **ResNet34-BEFB** | **14.88%** | 1.01% | **25.17%** | 2.98% |

**Table 13. Comparison of removing threshold layer and Sobel layers respectively.**

| | CIFAR-10 | | SVHN | |
|---|---|---|---|---|
| | FGSM | PGD | FGSM | PGD |
| VGG16-BEFB-tlre | 40.55% | 6.73% | 48.70% | 10.46% |
| VGG16-BEFB-slre | 33.73% | 2.90% | 55.22% | 15.89% |
| VGG16-BEFB | **45.60%** | **8.05%** | **59.28%** | **19.48%** |
| ResNet34-BEFB-tlre | 13.58% | 0.09% | 24.22% | 2.44% |
| ResNet34-BEFB-slre | 9.11% | 0.86% | 16.42% | 2.18% |
| ResNet34-BEFB | **14.88%** | **1.01%** | **25.17%** | **2.98%** |

## Acknowledgment

The authors want to thank Dr. Huo-Ping Yi for the insights on evaluating the robustness of DCNN.

## Author contributions

**Conceptualization:** Jin Ding.

**Data curation:** Jin Ding.

**Investigation:** Yong-Zhi Sun, Ping Tan.

**Methodology:** Jin Ding, Jie-Chao Zhao.

**Software:** Ji-En Ma.

**Supervision:** Ping Tan, Jia-Wei Wang, Ji-En Ma, You-Tong Fang.

**Validation:** Jie-Chao Zhao, Yong-Zhi Sun, Ping Tan, Jia-Wei Wang, Ji-En Ma.

**Visualization:** Jin Ding, Jie-Chao Zhao, Yong-Zhi Sun.

**Writing – original draft:** Jin Ding.

**Writing – review & editing:** Jin Ding.

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
