## [Decision Letter · Decision Letter 0]

8 Apr 2025

PONE-D-25-09891Learnable Edge Detectors Can Make Deep Convolutional Neural Networks More RobustPLOS ONE

Dear Dr. Ding,

Thank you for submitting your manuscript to PLOS ONE. After careful consideration, we feel that it has merit but does not fully meet PLOS ONE’s publication criteria as it currently stands. Therefore, we invite you to submit a revised version of the manuscript that addresses the points raised during the review process.

 

The reviewers noted that your idea shares similarities with prior work such as “DEA-Net.” Please cite this and clearly distinguish your contribution. The most recent references in your paper are from 2023. To position your work accurately within the current research landscape, please include and discuss more recent (2024–2025) papers in the field.Your method introduces an interesting idea with learnable edge features, but the theoretical explanation of how this improves robustness is quite limited. Expand your discussion on how the proposed edge detector helps maintain shape-related cues under adversarial attacks. More in-depth analysis and intuition would help readers understand why this strategy works.Several key training details are missing. Please specify the number of training epochs, optimizer used, learning rate(s), and any other relevant hyperparameters. Clarify how hyperparameters like the number of layers (l) and the coefficient (t) were chosen, especially why l varies across datasets. Describe the training pipeline: are both branches of your model trained jointly? Is the backbone trained from scratch or fine-tuned from a pretrained model? How does the inclusion of Sobel-like layers influence edge detection in earlier CNN layers?You're only reporting the best results out of five training runs. Please include averages and standard deviations to give a more accurate and reliable picture of your model’s performance. Clarify the meaning of bold values in Table 4. For Table 7, explain why your model performs significantly better under Gaussian noise compared to the baseline results in Table 3.The work lacks comparisons with more recent or strong baseline models. Please include evaluations against more state-of-the-art (SOTA) adversarially robust architectures.Consider comparing with other methods that incorporate shape-based or multi-branch architectures for robustness. Test your model with additional backbones beyond VGG16 and ResNet34 to demonstrate its general applicability.The current adversarial attacks used are somewhat limited. Please incorporate more challenging or recent attack methods to thoroughly evaluate the robustness of your approach. It would also be helpful to explain how you selected the parameters for generating adversarial perturbations.Please elaborate on why you chose to use the STE (Straight-Through Estimator) for training binarized edge features. Are there other alternatives, and if so, why was STE preferred?There’s concern that your edge detection approach only captures basic patterns (horizontal, vertical, diagonal). Could your model handle more complex or irregular edges? If so, please provide evidence or discussion to support this.For transparency and reproducibility, the reviewers request that you share your implementation—ideally by uploading the code to a public repository or including it as supplementary material.The overall quality of English needs improvement. Several sentences are unclear, and some transitions are abrupt. Please rewrite unclear sections and polish the writing for readability and flow, particularly when presenting your main ideas.

We look forward to receiving your revised manuscript.

Kind regards,

Panos Liatsis, PhD

Academic Editor

PLOS ONE

Additional Editor Comments (if provided):

Reviewers' comments:

Reviewer's Responses to Questions

**Comments to the Author**

1. Is the manuscript technically sound, and do the data support the conclusions?

Reviewer #1: Partly

Reviewer #2: Partly

Reviewer #3: Partly

2. Has the statistical analysis been performed appropriately and rigorously? 

Reviewer #1: Yes

Reviewer #2: No

Reviewer #3: N/A

3. Have the authors made all data underlying the findings in their manuscript fully available?

Reviewer #1: Yes

Reviewer #2: Yes

Reviewer #3: Yes

4. Is the manuscript presented in an intelligible fashion and written in standard English?

Reviewer #1: No

Reviewer #2: Yes

Reviewer #3: Yes

5. Review Comments to the Author

Reviewer #1: This work proposes a learnable edge detector to enhance the robustness of DCNNs. However, I have several major concerns regarding the article:

1. The English in the manuscript requires significant polishing. Some transitions are too abrupt, for example: "It prompts us that, enhancing the robustness of DCNNs can be achieved by using the binary edge features which can be seen as a kind of shape features. ", the motivation needs a deeper analysis.

2. The idea is similar with "DEA-Net: Single image dehazing based on detail-enhanced convolution and content-guided attention", typically, the learnable edge part. Needs proper citation.

3. The most recent reference cited in this work is from 2023. Given that it is now 2025, the paper would benefit from incorporating and comparing with more recent related works.

4. It seems that the proposed model not compare with SOTA models, please implement this.

5. If possible, please attach your code via file or opensource it.

6. The experimental setup lacks critical details. Information such as the number of training epochs, optimizer used, and learning rate settings should be explicitly provided

Reviewer #2: Publication is well written and presents an interesting idea of using learnable Sobel-like layers as an addition to two already established CNN architectures. Presented results showcase a good performance of a model under multiple adversarial attacks. However, authors include only top results from 5 trainings which does not provide an appropriate statistical analysis. Authors should provide an average across those 5 trainings and standard deviation or similar metric.

Please provide more information about model training in Experimental Settings section:

1. How number of layers l and coefficient t were chosen. Why l is different for only one dataset?

2. Are both branches of the model trained? Is it training from scratch or is the backbone somehow pretrained? If backbone is also trained, how are its weight influence? Usually, some layers in CNN learn to detect edges - is this somehow changed when there is explicit training of Sobel layers?

3. How perturbation parameters were selected?

Other questions:

1. Table 4: What do bold values mean?

2. Table 7: Why performance of the model on images with Gaussian noise is so much higher than the baseline one in Table 3?

Reviewer #3: Summary

This paper investigates the vulnerability of DCNNs to small adversarial perturbations and proposes a "learnable edge detector" with a binarized edge feature branch (BEFB). The effectiveness of the method is validated under both white-box and black-box attacks across multiple datasets.

Strengths

1.Innovative methodology: The edges with texture features, enabled by the four-directional Sobel-like kernels and threshold-based binarized edge features, demonstrates promising potential in enhancing adversarial robustness.

2.The proposed BEFB branch improves model robustness when integrated with backbone networks (VGG16, ResNet34) without significantly increasing computational overhead.

Weaknesses

1.The paper lacks thorough theoretical analysis of how the learnable edge detector functions. Although the authors provide visualizations of "clean vs. adversarial" edge features, the analysis of why these edge features help the model retain more shape-related cues is rather brief. A deeper discussion would help readers better understand how such structured edge features contribute to robustness against adversarial attacks.

2. Adversarial attacks are more limited, and while the authors tested traditional attack methods, they did not delve into the impact of more advanced adversarial methods. Additionally, comparisons with other shape-based or multi-branch robustness enhancement methods are lacking, making it difficult to fully demonstrate the superiority of the proposed approach.

3. Limited innovation and model generalizability. Although the designed convolutional kernels offer a novel approach to edge detection, they are primarily limited to fixed horizontal, vertical, and diagonal edges. The effectiveness of the method on more complex edges (e.g., irregular edges) is not thoroughly investigated. Thus, the method may be mainly suitable for basic edge detection tasks, while its capability to handle more complex edge features in real-world images remains unclear. Moreover, the authors used the STE approach, however this approach is more common when dealing with similar problems. As a result, innovativeness may be lacking.

Suggestions

1.The authors should include experiments with more sophisticated adversarial strategies to further test the proposed model.

2. Comparisons with additional backbone networks should be added to validate the method's performance across different attack strategies and network architectures.

3. It is recommended that the authors further justify the choice of this STE this method and explore whether other more effective alternatives exist.

6. PLOS authors have the option to publish the peer review history of their article (what does this mean?). If published, this will include your full peer review and any attached files.

Reviewer #1: No

Reviewer #2: No

Reviewer #3: No

---

## [Author Response · Author response to Decision Letter 1]

12 May 2025

Learnable Edge Detectors Can Make Deep Convolutional Neural Networks More Robust (PONE-D-25-09891)

We would like to express many thanks to reviewers and associate editor for their valuable comments, which help to enhance and improve the quality of manuscript considerably. We endeavored to address all the comments and our reflections are provided below point by point. An updated version of the paper being closed is modified based on the proposed comments. Now we summarize the responses to the reviewers’ comments as follows:

Reviewer 1:

Reviewer Comments:

The English in the manuscript requires significant polishing. Some transitions are too abrupt, for example: "It prompts us that, enhancing the robustness of DCNNs can be achieved by using the binary edge features which can be seen as a kind of shape features. ", the motivation needs a deeper analysis.

Authors Responses:

Thanks for your comments. A detailed explanation of the motivation is given at the beginning of the second paragraph of Section Introduction (marked in red).

We have proofread the manuscript and made several modifications, listed below:

[1] In the first line of Abstract and Section Introduction, “Deep convolutional neural networks (DCNN for short)” has been changed to “Deep convolutional neural networks (DCNNs)”.

[2] In the second line of Abstract, “Improving DCNN’s robustness” has been changed to “Improving DCNNs’ robustness”.

[3] In the first paragraph of Section Introduction, “which brings the potential hazards when applying DCNNs to the safety-critical applications” has been changed to “which poses potential hazards for safety-critical applications”.

[4] In the second paragraph of Section Introduction and the last paragraph of Section Proposed Approach, “learnt” has been changed to “learned”.

[5] In the second paragraph of Section Introduction, “STE technique” has been changed to “Straight-Through Estimator (STE) technique”.

[6] In the second paragraph of Section Related Work, “DCNNs are prone to be attacked” has been changed to “DCNNs are vulnerable to attacks”.

[7] In the second paragraph of Section Proposed Approach, “Inspired by the fact that shape feature is the main factor relied on by human beings to recognize objects” has been changed to “Inspired by the observation that humans primarily rely on shape features to recognize objects”.

[8] In the last paragraph of Section Proposed Approach, “Note that, the gradient of activation function of Eq. (5) is zero. To update the weights in BEFB, STE technique [14-16] is employed.” has been changed to “Note that the activation function in Eq. (5) has a zero gradient. Therefore, the STE technique [14-16] is used to update the weights in BEFB.”

[9] In the third paragraph of Section Experiments, “are on a par” has been changed to “perform comparably”.

[10] “BEFB integrated models” has been changed to “BEFB-integrated models”.

All these changes are marked in red.

Reviewer Comments:

The idea is similar with "DEA-Net: Single image dehazing based on detail-enhanced convolution and content-guided attention", typically, the learnable edge part. Needs proper citation.

Authors Responses:

Thanks for your comments. The paper “DEA-Net: Single image dehazing based on detail-enhanced convolution and content-guided attention” has been cited as Ref [15]. This paper proposed DEA-Net to perform image dehazing, part of which are four difference convolution kernels. These kernels are similar with the learnable edge detectors in our manuscript. Both can enhance the details learnt by DCNNs. The former used these enhanced details to dehaze images, and the latter leveraged these enhanced details to generate the binary shape-like features to improve robustness of DCNNs.

This paper “DEA-Net: Single image dehazing based on detail-enhanced convolution and content-guided attention” has been cited in the fourteenth line of the second paragraph in Section Introduction (marked in red).

Reviewer Comments:

The most recent reference cited in this work is from 2023. Given that it is now 2025, the paper would benefit from incorporating and comparing with more recent related works.

Authors Responses:

Thanks for your comments. The backbone models, attack methods, and robustness enhancement techniques employed in the manuscript for testing the performance of BEFB are widely used in DCNN-related research articles, e.g., ResNet34, AA, and AT. In the revised manuscript, we use WRN34-10 model, C&W attack method, BORT and DDPM robustness enhancement techniques to further test the performance of BEFB. The newly added references have been marked in red.

Reviewer Comments:

It seems that the proposed model not compare with SOTA models, please implement this.

Authors Responses:

Thanks for your comments. The performance of using the proposed BEFB to enhance robustness of DCNN is inferior than that of using SOTA robustness enhancement methods, like AT and PCL. However, when combining BEFB-integrated models with these SOTA methods, the robustness enhancement performance is better than that of combining original models with SOTA methods. The relevant experiments are conducted in Section Experiments--Combining BEFB-integrated models with SOTA robustness enhancement methods. Considering BEFB is lightweight and can be easily integrated with existing popular backbones, it depicts the superiority of BEFB. In the revised manuscript, we newly add two robustness enhancement techniques--DDPM and BORT, and the experimental results show the robustness enhancement performance of combining BEFB-integrated models with DDPM/BORT is better than that of combining original models with DDPM/BORT. For more details, please refer to Section Experiments--Combining BEFB-integrated models with SOTA robustness enhancement methods (marked in red).

Reviewer Comments:

If possible, please attach your code via file or opensource it.

Authors Responses:

Thanks for your comments. The project can be found in https://github.com/dingjin/BEFB.

Reviewer Comments:

The experimental setup lacks critical details. Information such as the number of training epochs, optimizer used, and learning rate settings should be explicitly provided.

Authors Responses:

Thanks for your comments. The number of training epochs is 50000. The optimizer used in the experiments is the SGD optimizer, with a momentum of 0.9 and a weight decay coefficient of 5e-4. The initial learning rate is set to 0.01, and the batch size for training samples is set to 128. Section Experiments-- Experimental Settings has been modified accordingly (marked in red).

Reviewer 2:

Reviewer Comments:

However, authors include only top results from 5 trainings which does not provide an appropriate statistical analysis. Authors should provide an average across those 5 trainings and standard deviation or similar metric.

Authors Responses:

Thanks for your comments. The values of training metrics in Table 3 have been modified to the average across 5 trainings. And the corresponding standard deviations have been provided in Table 4. Section Experiments-- Effects of BEFB on model training has been modified (marked in red).

Reviewer Comments:

How number of layers l and coefficient t were chosen. Why l is different for only one dataset?

Authors Responses:

Thanks for your comments. The number of layers l and coefficient t are chosen empirically. The Table R1 below shows the classification accuracy of BEFB-integrated models under AA attack on CIFAR-10 dataset for different l with t fixed to 0.8. The Table R2 below shows the classification accuracy of BEFB-integrated models under AA attack on CIFAR-10 dataset for different t with l fixed to 2.

Table R1 Performance comparison for different l with the fixed t = 0.8

l=1 l=2 l=3 l=4

VGG16-BEFB 13.78% 14.57% 14.01% 12.72%

ResNet34-BEFB 14.46% 15.04% 14.73% 14.03%

Table R2 Performance comparison for different t with the fixed l = 2

t=0.2 t=0.4 t=0.6 t=0.8 t=0.9

VGG16-BEFB 13.70% 13.65% 14.06% 14.57% 13.99%

ResNet34-BEFB 14.11% 14.30% 15.83% 15.04% 14.07%

Reviewer Comments:

Are both branches of the model trained? Is it training from scratch or is the backbone somehow pretrained? If backbone is also trained, how are its weight influence? Usually, some layers in CNN learn to detect edges - is this somehow changed when there is explicit training of Sobel layers?

Authors Responses:

Thanks for your comments. Yes. Both branches of the model are trained. Both are trained from scratch. The backbone is trained to output traditional texture features. BEFB is trained to output the binary edge features. When there is explicit training of Sobel layers, some layers in backbone we think can also learn to detect edges as common CNNs do. Fig. 8-10 demonstrate the gray features output by backbone. It can be observed edge information has been learned.

Reviewer Comments:

How perturbation parameters were selected?

Authors Responses:

Thanks for your comments. The selected perturbation parameters are commonly used in the publications on testing adversarial robustness of CNNs.

Reviewer Comments:

Table 4: What do bold values mean?

Authors Responses:

Thanks for your comments. Since the gradient of the threshold function is zero, we utilize the STE to ensure weight updates can proceed. However, when generating adversarial examples (AE) using white-box attack algorithms (such as FGSM), STE can bring obfuscated gradient effect [19-21], which means the generated AEs are not powerful enough, leading to an artificially high classification accuracy for the BEFB-integrated models. Therefore, this paper tested two alternatives to STE: the sigmoid function and zero gradient to generate AEs. The experimental results in Table 5 (Table 4 in original manuscript) show that zero gradient in most cases can allow FGSM attack to achieve lower classification accuracy of BEFB-integrated models. Consequently, we choose zero gradient for generating AEs.

Reviewer Comments:

Table 7: Why performance of the model on images with Gaussian noise is so much higher than the baseline one in Table 3?

Authors Responses:

Thanks for your comments. Adversarial examples generated with Gaussian noise do not utilize the model's gradient information, therefore the model's classification accuracy is higher. In contrast, the attack methods in Table 6 (Table 5 in original manuscript) are white-box attacks, which leverage the model's gradient information, resulting in lower classification accuracy for the model. Results in both tables show the BEFB-integrated models can have better robustness performance than the original models.

Reviewer 3:

Reviewer Comments:

The paper lacks thorough theoretical analysis of how the learnable edge detector functions. Although the authors provide visualizations of "clean vs. adversarial" edge features, the analysis of why these edge features help the model retain more shape-related cues is rather brief. A deeper discussion would help readers better understand how such structured edge features contribute to robustness against adversarial attacks.

Authors Responses:

Thanks for your comments. How the learnable edge detectors help improve the robustness of DCNNs is depicted in Fig. 7-10/Fig. 11-14. In these figures, the gray features are obtained from backbone branch, and the binary features are obtained from BEFB. It can be observed in Fig. 7 that, using the original VGG16 model, the difference of the extracted gray features between clean examples and adversarial examples (AE) is large, which makes AE prediction results all wrong under FGSM ϵ=16 attack. While in Fig. 8, using BEFB-integrated VGG16 model, the difference of the extracted binary and gray features between clean examples and AEs is small, which keeps AE prediction results all right under FGSM ϵ=16 attack. Fig. 9 and Fig. 10 add 4 more and 8 more perturbations to evaluate the performance of BEFB-integrated VGG16 model, respectively. It can be observed in both figures that, the difference of the extracted binary features between clean examples and AEs is small, demonstrating the extracted binary features from BEFB are less susceptible to the noise. Similarly, Fig. 11-14 demonstrate how the learnable edge detectors help improve the robustness of ResNet34 model. For more details, please refer to the fourth paragraph and the fifth paragraph of Section Experiments--Performance comparison between BEFB-integrated models and original models under attacks.

Reviewer Comments:

Adversarial attacks are more limited, and while the authors tested traditional attack methods, they did not delve into the impact of more advanced adversarial methods. Additionally, comparisons with other shape-based or multi-branch robustness enhancement methods are lacking, making it difficult to fully demonstrate the superiority of the proposed approach.

Authors Responses:

Thanks for your comments. In the revised manuscript, C&W attack method is included to further test the performance of the proposed BEFB. And the experimental results in Table 7 show the BEFB-integrated models perform better than original models under C&W attack on four datasets.

The performance of using the proposed BEFB to enhance robustness of DCNN is inferior than that of using SOTA robustness enhancement methods, like AT and PCL. However, when combining BEFB-integrated models with these SOTA methods, the robustness enhancement performance is better than that of combining original models with SOTA methods. The relevant experiments are conducted in Section Experiments--Combining BEFB-integrated models with SOTA robustness enhancement methods. Considering BEFB is lightweight and can be easily integrated with existing popular backbones, it depicts the superiority of BEFB.

The shape-based robustness enhancement methods require datasets of pixel-level segmentation labeling. Therefore, in the revised manuscript, we add a multi-branch robustness enhancement technique--BORT, and a SOTA robustness enhancement technique—DDPM for comparison. The experimental in Table 11 results show the robustness enhancement performance of combining BEFB-integrated model with BORT/DDPM is better than that of combining original model with BORT/DDPM. For more details, please refer to Section Experiments--Combining BEFB-integrated models with SOTA robustness enhancement methods (marked in red).

Reviewer Comments:

Limited innovation and model generalizability. Although the designed convolutional kernels offer a novel approach to edge detection, they are primarily limited to fixed horizontal, vertical, and diagonal edges. The effectiveness of the method on more complex edges (e.g., irregular edges) is not thoroughly investigated. Thus, the method may be mainly suitable for basic edge detection tasks, while its capability to handle more complex edge features in real-world images remains unclear. Moreover, the authors used the STE approach, however this approach is more common when dealing with similar problems. As a result, innovativeness may be lacking.

Authors Responses:

Thanks for your comments. The four learnable Sobel-like edge detectors are designed for detecting horizontal, vertical, positive diagonal, negative diagonal edges, which means in these directions, the detection output will be relatively strong. In other directions, these edge detectors can also work, and the detection output will be a little weaker.

When the derivative of the neural network's activation function does not exist or is zero, to the best of our knowledge, the STE is a commonly-used and effective technique to make weights can be updated. It sets the derivative to a constant value to facilitate training. In BEFB, the activation function of the threshold layer is a threshold function, and we utilize STE to enable errors can be back-propagated to the Sobel layers for weights updating. The experimental results demonstrate using STE technique, the trained BEFB-integrated models have superior performance under various attacks.

The main innovation of this paper lies in proposing BEFB to learn shape-like features to enhance the robustness of deep convolutional neural networks.

Reviewer Comments:

The authors should include experiments with more sophisticated adversarial strategies to further test the proposed model.

Authors Responses:

Thanks f

---

## [Decision Letter · Decision Letter 1]

9 Jun 2025

PONE-D-25-09891R1Learnable Edge Detectors Can Make Deep Convolutional Neural Networks More RobustPLOS ONE

Dear Dr. Ding,

Thank you for submitting your manuscript to PLOS ONE. After careful consideration, we feel that it has merit but does not fully meet PLOS ONE’s publication criteria as it currently stands. Therefore, we invite you to submit a revised version of the manuscript that addresses the points raised during the review process.

Please resolve any ambiguities between the manuscript text and figuresPlease include and analyze more recent references (from 2024) in the state-of-the-art

We look forward to receiving your revised manuscript.

Kind regards,

Panos Liatsis, PhD

Academic Editor

PLOS ONE

Journal Requirements:

Reviewers' comments:

Reviewer's Responses to Questions

**Comments to the Author**

1. If the authors have adequately addressed your comments raised in a previous round of review and you feel that this manuscript is now acceptable for publication, you may indicate that here to bypass the “Comments to the Author” section, enter your conflict of interest statement in the “Confidential to Editor” section, and submit your "Accept" recommendation.

Reviewer #1: (No Response)

Reviewer #2: All comments have been addressed

2. Is the manuscript technically sound, and do the data support the conclusions?

Reviewer #1: Yes

Reviewer #2: Yes

3. Has the statistical analysis been performed appropriately and rigorously? 

Reviewer #1: N/A

Reviewer #2: Yes

4. Have the authors made all data underlying the findings in their manuscript fully available?

Reviewer #1: No

Reviewer #2: Yes

5. Is the manuscript presented in an intelligible fashion and written in standard English?

Reviewer #1: Yes

Reviewer #2: Yes

6. Review Comments to the Author

Reviewer #1: Thanks for the response, however, I do think there exist a minor issues:

- There is a discrepancy in the paper regarding training epochs. While it claims "The number of training epochs is set to 50000," Figure 6 shows convergence after only 60 epochs. This inconsistency needs clarification from the authors.

- Including only one or two papers from 2024 is insufficient for a 2025 publication. At least 10% of references should come from recent work (2024-2025). A more comprehensive analysis of recent related work is necessary.

Reviewer #2: Authors addressed comments in satisfactory way and work is acceptable to be published.

7. PLOS authors have the option to publish the peer review history of their article (what does this mean?). If published, this will include your full peer review and any attached files.

Reviewer #1: No

Reviewer #2: No

---

## [Author Response · Author response to Decision Letter 2]

25 Jun 2025

Learnable Edge Detectors Can Make Deep Convolutional Neural Networks More Robust (PONE-D-25-09891R1)

We would like to express many thanks to reviewers and associate editor for their valuable comments, which help to enhance and improve the quality of manuscript considerably. We endeavored to address all the comments and our reflections are provided below point by point. An updated version of the paper being closed is modified based on the proposed comments. Now we summarize the responses to the reviewers’ comments as follows:

Reviewer 1:

Reviewer Comments:

There is a discrepancy in the paper regarding training epochs. While it claims "The number of training epochs is set to 50000," Figure 6 shows convergence after only 60 epochs. This inconsistency needs clarification from the authors.

Authors Responses:

Thanks for your comments. In Figure 5 and Figure 6, all four models were trained on the CIFAR-10 dataset for 50000 epochs. To more clearly compare the training dynamics curves between BEFB-integrated models and original models, Figures 5 and 6 illustrate the changes in training loss and accuracy for the first 60 epochs. As can be seen from the figures, the BEFB-integrated models and the original models exhibit similar converging speed and accuracy. Section “Experiments--Effects of BEFB on model training” has been modified to clarify this inconsistency (marked in red).

Reviewer Comments:

Including only one or two papers from 2024 is insufficient for a 2025 publication. At least 10% of references should come from recent work (2024-2025). A more comprehensive analysis of recent related work is necessary.

Authors Responses:

Thanks for your comments. Five references from 2024 to 2025 have been added and analyzed in Section “Related Work” (marked in red).

Reviewer 2:

Reviewer Comments:

Authors addressed comments in satisfactory way and work is acceptable to be published.

Authors Responses:

Thanks for your efforts and time.

Finally, we would thank all reviewers again for the valuable and critical comments to make the paper more understandable and clearer.

Sincerely,

The Authors.

---

## [Editor Report · Decision Letter 2]

30 Jul 2025

Learnable Edge Detectors Can Make Deep Convolutional Neural Networks More Robust

PONE-D-25-09891R2

Dear Dr. Ding,

We’re pleased to inform you that your manuscript has been judged scientifically suitable for publication and will be formally accepted for publication once it meets all outstanding technical requirements.

Kind regards,

Panos Liatsis, PhD

Academic Editor

PLOS ONE
---

## [Editor Report · Acceptance letter]

PONE-D-25-09891R2

PLOS ONE

Dear Dr. Ding,

I'm pleased to inform you that your manuscript has been deemed suitable for publication in PLOS ONE. Congratulations! Your manuscript is now being handed over to our production team.

Kind regards,

on behalf of

Professor Panos Liatsis

Academic Editor

PLOS ONE